# SQL-R1: Training Natural Language to SQL Reasoning Model By Reinforcement Learning

**Peixian Ma**[1,2], **Xialie Zhuang**[1,3], **Chengjin Xu**[1,4,∗], **Xuhui Jiang**[1,4], **Ran Chen**[1], **Jian Guo**[1]
[1]IDEA Research, International Digital Economy Academy
[2]The Hong Kong University of Science and Technology (Guangzhou)
[3]University of Chinese Academy of Sciences
[4]DataArc Tech Ltd.
pma929@connect.hkust-gz.edu.cn, xuchengjin@idea.edu.cn

## Abstract

Natural Language to SQL (NL2SQL) enables intuitive interactions with databases by transforming natural language queries into structured SQL statements. Despite recent advancements in enhancing human-computer interaction within database applications, significant challenges persist, particularly regarding the reasoning performance in complex scenarios involving multi-table joins and nested queries. Current methodologies primarily utilize supervised fine-tuning (SFT) to train the NL2SQL model, which may limit adaptability and interpretability in new environments (e.g., finance and healthcare). In order to enhance the reasoning performance of the NL2SQL model in the above complex situations, we introduce SQL-R1, a novel NL2SQL reasoning model trained by the reinforcement learning (RL) algorithms. We design a specialized RL-based reward function tailored for NL2SQL tasks and discussed the impact of cold start and synthetic data on the effectiveness of intensive training. In addition, we achieve competitive accuracy using only a tiny amount of synthetic NL2SQL data for augmented training and further explore data engineering for RL. In existing experiments, SQL-R1 achieves execution accuracy of 88.6% and 67.1% on the benchmark Spider and BIRD, respectively. The code is available at https://github.com/IDEA-FinAI/SQL-R1.

## 1 Introduction

Natural Language to SQL (NL2SQL, or Text2SQL) converts natural language questions (NL) into structured SQL statements, simplifying database interaction without requiring database expertise [1, 2]. Recent advancements in NL2SQL have significantly enhanced the level of human-computer interaction within database query applications and contribute to a wide range of data science analysis tasks [3, 4]. Current NL2SQL models mainly focus on optimizing workflows and their components, such as schema linking [5, 6], content retrieval [7], generation correction [8–12].

Despite these advancements, improving the NL2SQL system's reasoning performance in complex database scenarios remains a considerable challenge. As shown in Figure 1, schema complexity may lead to generation errors in processing multi-table joins and nested queries, and it is difficult for individually trained models to think and process complex semantics. Currently, a significant portion of NL2SQL research is devoted to training open-source large language models (LLMs) by supervised fine-tuning (SFT) [13–15] to achieve accuracy at a smaller model scale compared to approaches using closed-source LLMs (e.g., GPT-4, GPT-4o) [8, 10, 16]. However, SFT relies on the database schema structure and the training data scale. This may lead to the existing model's instability in domain adaptation and generalization in new database environments. Additionally, the lack of interpretability

---

∗Corresponding author.

39th Conference on Neural Information Processing Systems (NeurIPS 2025).

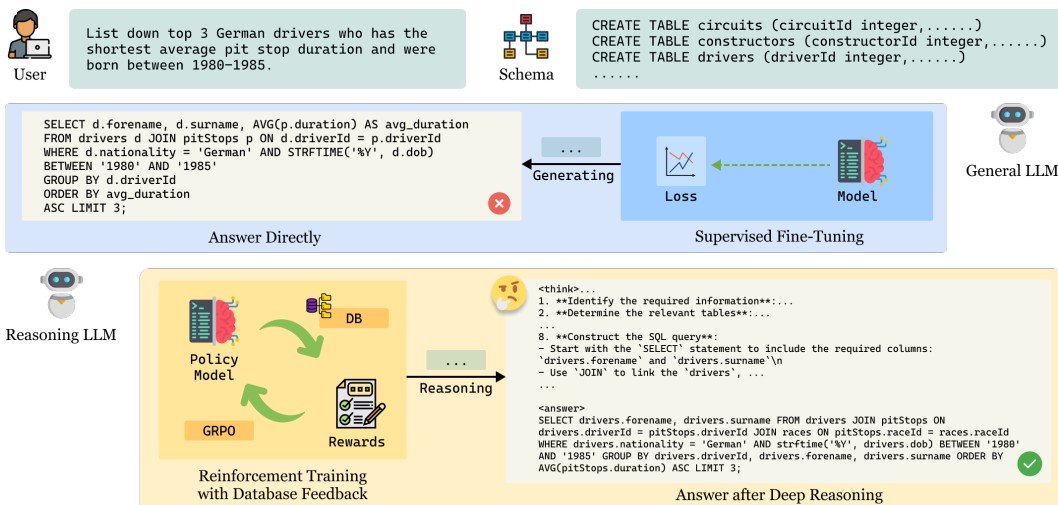

Figure 1: Demonstration of our work. Previous work on NL2SQL primarily relies on supervised fine-tuning to enable the model to learn how to generate SQL. However, in the case of complex database schema or ambiguous semantics, the fine-tuned model may struggle to produce SQL that does not align with the user's intentions, as it depends on a fixed generation strategy and previous data. By introducing reinforcement learning algorithms, the model can receive intuitive feedback from the database during the training process. This feedback encourages the model to independently explore various SQL generation reasoning approaches, ultimately enhancing the accuracy of its output.

of NL2SQL reasoning logic limits the application of the model in high-risk fields, such as finance and healthcare.

Recently, reinforcement learning (RL) has shown great potential in training the reasoning ability of LLMs in recent research. Compared with supervised fine-tuning, reinforcement learning can dynamically adjust the decision-making strategy of the LLMs through interaction with the environment, thereby achieving superior performance in complex reasoning tasks [17]. RL-based methods have proven effective in enhancing model reasoning and generalization capabilities in financial reasoning [18], search engines [19], and mathematical reasoning [20, 21].

Based on the above inspiration, we proposed SQL-R1, a NL2SQL reasoning model trained by the reinforcement learning algorithm. Figure 1 demonstrates the overview of our work. In the following sections, we will focus on answering the following critical questions:

***Q1****: Can we design a specific reinforcement learning algorithm for the NL2SQL task and successfully train a NL2SQL reasoning model?* In contrast to SFT, RL algorithms prioritize the direct optimization of NL2SQL reasoning, specifically by generating SQL queries that accurately reflect the user's query intent. The design of effective feedback mechanisms for reinforcement learning presents a substantial challenge in developing NL2SQL reasoning models. Appropriately structured rewards within the reinforcement learning framework can significantly enhance its performance.

***Q2****: For the RL-based NL2SQL reasoning model, do we need to perform a specific form of cold start on it?* For the existing base model, an effective cold start can strengthen the model's instruction-following ability and activate its NL2SQL generation ability, thus promoting it to generate higher-quality SQL queries in reinforcement learning exploration. Designing the form of a cold start will also be a significant challenge.

***Q3****: Can we deploy sustainable data engineering for training robust and efficient NL2SQL reasoning models?* RL training relies on high-quality training data, while current NL2SQL tasks lack a large amount of real data for training. How to develop the data support for NL2SQL reasoning model based on the existing data engineering technology is an important challenge to solve the model training, improve the robustness and generalization of the model.

Above all, the contribution of this work are as follows:

- **Explicit NL2SQL Reasoning Model:** We propose SQL-R1, a NL2SQL reasoning model trained on a few NL2SQL data (e.g., 5K) currently, which can achieve 88.6% and 66.6% accuracy on the leading benchmark Spider-Test and BIRD, respectively, and can output a detailed explicit reasoning process.

- **Training Strategy for NL2SQL Reasoning Model:** We extensively explored the impact of cold-start training on SQL-R1, developing a training strategy that integrates SFT and RL. Our findings highlight the strategy of using synthetic data to enhance model performance and robustness, offering key insights into optimizing NL2SQL reasoning model training.

## 2    SQL-R1

### 2.1    Overview

This section mainly introduces two forms of training NL2SQL models via RL algorithms: direct reinforcement training and reinforcement training via cold start after training. Among them, cold start refers to using specific data to train the base model by SFT first so that it has a particular ability to think and follow instructions. In addition, due to the limited real data, we use the latest synthetic data to support the above training process. Section 2.2 will introduce our current data engineering solution, Section 2.3 will introduce the SFT algorithm and the RL algorithm designed for NL2SQL.

### 2.2    Data Preparation

#### 2.2.1    Source

Currently, we utilize the SynSQL-2.5M [22] dataset as primary data source, which is the first million-scale synthetic NL2SQL dataset, encompassing over 2.5 million diverse and high-quality data samples. Each sample consists of a quadruple comprising a database, a natural language question, an SQL query, and a chain-of-thought (CoT) solution. The dataset features more than 16,000 synthetic databases across various domains, thereby ensuring extensive coverage of realistic scenarios. SynSQL-2.5M includes a wide range of SQL complexity levels, from simple single-table queries to intricate multi-table joins, functions, and common table expressions.

#### 2.2.2    Preprocessing

**SFT Dataset.**    In this study, we investigated the impact of the cold start condition in SFT on RL training. Currently, we utilized a dataset comprising 200,000 samples drawn from the SynSQL-2.5M for the SFT training, whose sample size is uniform across different difficulty levels, with each level comprising 50000 samples. For clarity, we will refer to this subset as **SynSQL-200K** in subsequent sections. It is essential to highlight that the query results obtained from the all SQL ground truth are exclusively are non-null values. For each sample $v = (x, t, y^*)$ in the SFT dataset $V$, $x$ represents the NL, while $t$ represents the reasoning process enclosed in <think>...</think> tags and $y^*$ denotes the SQL enclosed in <answer>...</answer> tags.

**RL Dataset.**    The current NL2SQL base model has demonstrated a strong capability in generating simple to moderate SQL queries. However, it exhibits limitations when tasked with the creation of more sophisticated SQL queries. Consequently, employing a dataset comprised of more challenging samples during the training process may prove beneficial in addressing these deficiencies and enhancing the model's overall performance in generating complex SQL. We randomly sampled 5K NL-SQL pairs from SynSQL-2.5M, whose complexity are **Complex**. For each NL-SQL pair $v = (x, y^*)$ in the RL dataset $V$, $x$ represents the NL, while $y^*$ denotes the SQL candidate generated by the model. The aim of reinforcement learning is to enhance the accuracy of the answers and ensure that they adhere to the expected format. The RL Dataset is introduced as **SynSQL-Complex-5K** in the next section. Notably, The input of the RL dataset does not include the CoT data of the original SynSQL-2.5M.

### 2.3 Training

#### 2.3.1 Supervised Fine-Tuning

In this study, we conduct SFT on the Qwen2.5-Coder-7B-Instruct model to enhance the model's capacity for instruction adherence and generation within the NL2SQL domain. We investigate two distinct strategies for SFT cold start training. The first one employs raw instructions focusing exclusively on SQL generation. We leverage the existing OmniSQL-7B [22] checkpoint for the reference. The second strategy utilizes full fine-tuning and reasoning generation instructions promoting the development of compliant thought processes alongside final answers.

#### 2.3.2 Reinforcement Training

In the reinforcement learning phase, we employ the Group Relative Policy Optimization (GRPO) algorithm to enhance our training protocol, which obviates the need for the value model, operates with less memory requirements, and facilitates a clear definition of reward targets, rendering it an optimal choice for the effective optimization of the NL2SQL policy model [23].

For each natural language question aligned with its corresponding database schema, the policy model generates a set of $G$ SQL candidates $\{o_1, o_2..., o_G\}$ from the old policy $\pi_{old}$, which are meticulously evaluated using a composite reward function that assigns specific reward scores. By concentrating on the relative performance of the SQL candidates within the group, GRPO adeptly calculates the rewards for each output, thereby guiding the policy update in accordance with our established objectives.

$$
\begin{aligned}
\mathcal{J}_{\text{GRPO}}(\theta) = & \mathbb{E}_{\mathbf{v} \sim P(\mathbf{V}), \{o_i\}_{i=1}^{G} \sim \pi_{\theta_{\text{old}}}(O|\mathbf{v})} \\
& \left[ \frac{1}{G} \sum_{i=1}^{G} \left( \min \left( r_i^{\text{ratio}} A_i, \text{clip} \left( r_i^{\text{ratio}}, 1 - \epsilon, 1 + \epsilon \right) A_i \right) - \beta D_{\text{KL}}(\pi_\theta \| \pi_{\text{ref}}) \right) \right],
\end{aligned} \quad (1)
$$

where $r_i^{\text{ratio}} = \frac{\pi_\theta(o_i|V)}{\pi_{old}(o_i|V)}$ represents the importance sampling ratio that quantifies the relative likelihood of generating output oi under the new policy $\pi_\theta$ compared to $\pi_{old}$; $A_i$ represents the group-relative advantage for each output; the clipping operator, hyperparameter $\epsilon$ and $\beta$ control the update step and divergence regularization; $\pi_{\text{ref}}$ represents the reference policy.

#### 2.3.3 Reward Function Design

When training NL2SQL reward model using reinforcement learning, we utilize a progressive feedback mechanism that consists of four types of rewards: Format Reward, Execution Reward, Result Reward, and Length Reward. This layered approach enhances the model's learning by providing detailed feedback at various stages.

**Format Reward.** We encourage the model to enclose the NL2SQL reasoning process within `<think>...</think>` tags and to present the final answer enclosed within `<answer>...</answer>` tags. To achieve a more standardized format for SQL queries, it is essential that all SQL statements be contained within ```sql...``` tags; failure to do so will result in their format as erroneous. The structure of the format reward function is delineated as follows:

$$
S_f = \begin{cases} 1, & \text{if format is correct} \\ -1, & \text{if format is incorrect} \end{cases} \quad (2)
$$

**Execution Reward.** Execution rewards are designed to evaluate the syntactic correctness of SQL candidates, preventing the model from generating messy, unexecutable responses. When the SQL candidate fails to execute correctly, the model will not receive all subsequent rewards. In addition, we limit the execution time to prevent the model from tending to generate too complex SQL for high rewards.

$$S_e = \begin{cases} 2, & \text{if SQL candidate is executable} \\ 0, & \text{if format is incorrect} \\ -2, & \text{if SQL candidate is not executable} \end{cases} \tag{3}$$

**Result Reward.** The accuracy of query results is an important criterion in the evaluation of SQL candidates. We prioritize the Result Reward as a key component of the reward mechanism, aimed at motivating the model to generate SQL candidates that aligns with the real intention of the user. To evaluate the correctness of SQL candidate query results and the associated feedback, we utilize Execution Accuracy (EX). In cases of incorrect results, we impose stringent penalties to guide the model in its subsequent reasoning.

$$S_r = \begin{cases} 3, & \text{if query result is correct} \\ 0, & \text{if format is incorrect or SQL candidate is not executable} \\ -3, & \text{if query result is incorrect} \end{cases} \tag{4}$$

**Length Reward.** We apply length reward mechanism to incentivize the model to produce more comprehensive reasoning process. It is divided into two components: The first component allocates half of the reward based on the proportional relationship between the total length of the answer and the maximum length of response; The second component computes the remaining half of the reward based on the ratio of the SQL candidate length within the `<answer>`, which aims to mitigate the occurrence of superfluous explanations in the response. When the response exceeds the maximum length, penalized feedback is given to the model.

$$S_l = \begin{cases} 0.5 \times S_{tl} + S_{al}, & \text{if query result is correct and } len_{response} <= \texttt{MAX LENGTH} \\ 0.5 + S_{al}, & \text{if query result is correct and } len_{response} > \texttt{MAX LENGTH} \\ 0, & \text{other cases} \end{cases} \tag{5}$$

where $S_{tl} = (len_{think} + len_{answer})\,/\,\texttt{MAX LENGTH}$ and $S_{al} = len_{sql}\,/\,len_{answer}$.

### 2.4 SQL Candidates Selection

In order to select the most appropriate SQL in the reasoning process, the model generates several SQL candidates and their thought processes for a problem. We execute all SQL candidates and select the SQL with the highest score as the final answer based on self-consistency voting. Notably, the reasoning response of SQL-R1 comprises an observable process of thinking and interpreting, making the results easier for the user to understand.

## 3 Experiments

### 3.1 Setup

**Evaluation Benchmark.** We evaluated the proposed SQL-R1 and related NL2SQL models on two benchmarks, Spider [24] and BIRD [25]. Spider comprises 10,181 questions paired with 5,693 complex SQL queries from 200 databases and 138 domains. BIRD comprises 12,751 NL2SQL pairs encompassing 95 databases from 37 specialized domains.

**Metric.** For fair comparisons, we follow the standard evaluation metric in previous works. We use Execution Accuracy (EX) the evaluation metric on Spider and BIRD benchmark. EX serves to estimate the proportion of questions that produce consistent outcomes for both the given query and its corresponding basic fact query across all query requests.

**Implementation Settings.** Currently, SQL-R1 is mainly built on Qwen2.5-Coder series models [26]. For the SFT training, we set the learning rate as 5e-5; batch size as 1. For the RL training, we set the

Table 1: Execution accuracy (%) of different NL2SQL methods on Spider and BIRD benchmark.

| NL2SQL Method | Base Model | Candidate Selection | Spider (Dev) | Spider (Test) | BIRD (Dev) |
|---|---|---|---|---|---|
| CodeS [14] | CodeS-15B | - | 84.9 | 79.4 | 57.0 |
| DTS-SQL [13] | Deepseek-Coder-7B | - | 85.5 | 84.4 | 55.8 |
| CHESS [7] | Deepseek-Coder-33B | - | - | 87.2 | 61.5 |
| Alpha-SQL [29] | Qwen2.5-Coder-7B | Self-Consistency | 84.0 | - | 66.8 |
| SQL-o1 [30] | Qwen2.5-Coder-7B | Self-Consistency | 84.7 | 85.1 | 66.7 |
| OmniSQL [22] | Qwen2.5-Coder-7B | Self-Consistency | 85.5 | 88.9 | 66.1 |
| DeepRetrieval [31] | Qwen2.5-Coder-7B | - | - | 76.1 | 56.0 |
| Reasoning-SQL [32] | Qwen2.5-Coder-14B | Self-Consistency | 81.4 | - | 65.3 |
| C3-SQL [10] | GPT-3.5-Turbo | Self-Consistency | 82.0 | 82.3 | - |
| DIN-SQL [8] | GPT-4 | - | 82.8 | 85.3 | - |
| DAIL-SQL [16] | GPT-4 | Self-Consistency | 83.6 | 86.2 | 54.8 |
| MAC-SQL [9] | GPT-4 | Self-Consistency | 86.8 | 82.8 | 59.4 |
| SuperSQL [33] | GPT-4 | Self-Consistency | 84.0 | 87.0 | 58.5 |
| MCTS-SQL [34] | GPT-4o | - | 88.7 | 86.6 | 69.4 |
| OpenSearch-SQL [35] | GPT-4o | Self-Consistency | - | 87.1 | 69.3 |
| CHASE-SQL [36] | Gemimi-1.5-Pro | - | - | 87.6 | 73.0 |
| **SQL-R1 (Ours)** | **Qwen2.5-Coder-3B** | **Self-Consistency** | **78.1** | **78.9** | **54.6** |
| **SQL-R1 (Ours)** | **Qwen2.5-Coder-7B** | **Self-Consistency** | **87.6** | **88.7** | **66.6** |
| **SQL-R1 (Ours)** | **Qwen2.5-Coder-14B** | **Self-Consistency** | **86.7** | **88.1** | **67.1** |

learning rate as 3e-7, rollout of actor model as 8; max response length as 2048. For inference, we set the count of SQL candidates as 8 and the temperature as 0.8.

For all NL2SQL data samples in the dataset, we first convert them into suitable input-output sequence pairs for training. Specifically, the input sequence includes natural language questions and related database schemas. Inspired by previous research [7, 22, 27, 28], the database schema was formatted as a CREATE TABLE statement with supplementary annotations including column attribute descriptions and representative values. Currently, annotations of representative values will not be added during the training phase for the time being to enhance the model's exploration ability during the reinforcement learning phase.

**Environment.** All experiments conducted in this study are performed on a server operating under the Ubuntu 20.04 Linux distribution. This server is equipped with Intel(R) Xeon(R) Platinum 8358 CPU @ 2.60 GHz CPU, and is complemented by 512 GB of system memory. The environment for training open-source LLMs comprises a configuration of 8 GPUs, each possessing 80 GB of memory and delivering a performance capacity of 312 TFLOPS when utilizing BF16 precision.

## 3.2 Main Results

**Performance on Main Benchmarks.** The results presented in Table 1 underscore the performance of SQL-R1 across various foundational models. Specifically, when trained on the Qwen2.5-Coder-3B, SQL-R1 achieved an execution accuracy of 78.1% on the Spider development dataset, 78.9% on the Spider test dataset, and 54.6% on the BIRD development dataset, demonstrating robust performance even with a smaller base model. Moreover, SQL-R1 exhibits a marked enhancement in accuracy when utilized with larger models; for instance, the Qwen2.5-Coder-7B model yielded accuracies of 87.6%, 88.7%, and 63.1% across the same Spider development, Spider test, and BIRD development datasets, respectively. The deployment of the even larger Qwen2.5-Coder-14B model further affirmed SQL-R1's capabilities, achieving high accuracy rates of 86.7%, 88.1%, and 67.1% on these datasets. In comparison to other solutions leveraging closed-source models, such as GPT-4 and GPT-4o, SQL-R1 demonstrates competitive and often superior performance. Notably, In comparison to contemporaneous exploration of reasoning-based NL2SQL (e.g., DeepRetrieval [31] and Reasoning-SQL [32]), SQL-R1 surpasses them on the Spider-Dev and BIRD-Dev datasets. This

Table 2: Execution accuracy (%) on Spider-DK, Spider-Syn, Spider-Realistic benchmark.

| NL2SQL Method | Base Model | Spider-DK | Spider-Syn | Spider-Realistic |
|:---:|:---:|:---:|:---:|:---:|
| SENSE [37] | CodeLlama-7B | 77.9 | 72.6 | 82.7 |
| ROUTE [38] | Llama3-8B | 74.6 | 77.4 | 80.9 |
| SQL-o1 [30] | Llama3-8B | 78.7 | 72.6 | 82.7 |
| OmniSQL [22] | Qwen2.5-Coder-7B | 77.8 | 69.6 | 78.0 |
| SQL-PaLM [39] | PaLM-2 | 67.5 | 70.9 | 77.4 |
| PURPLE [40] | GPT-4 | 75.3 | 74.0 | 79.9 |
| **SQL-R1 (Ours)** | **Qwen2.5-Coder-3B** | **70.5** | **66.4** | **71.5** |
| **SQL-R1 (Ours)** | **Qwen2.5-Coder-7B** | **78.1** | **76.7** | **83.3** |
| **SQL-R1 (Ours)** | **Qwen2.5-Coder-14B** | **79.3** | **78.5** | **86.2** |

Table 3: Execution accuracy (%) of different complexity levels on BIRD-Dev dataset.

| NL2SQL Method | Base Model | Simple | Moderate | Challenging | All |
|:---:|:---:|:---:|:---:|:---:|:---:|
| CodeS [14] | Codes-15B | 65.8 | 48.8 | 42.4 | 58.5 |
| DAIL-SQL [16] | GPT-4 | 63.0 | 45.6 | 43.1 | 55.9 |
| SuperSQL [33] | GPT-4 | 66.9 | 46.5 | 43.8 | 58.5 |
| **SQL-R1 (Ours)** | **Qwen2.5-Coder-7B** | **72.1** | **60.8** | **51.0** | **66.6** |
| **SQL-R1 (Ours)** | **Qwen2.5-Coder-14B** | **72.4** | **59.7** | **56.5** | **67.1** |

performance highlights the efficacy of our proposed approach in utilizing diverse base models to attain state-of-the-art results on complex NL2SQL reasoning tasks.

Table 2 demonstrates the comparison of execution accuracy on Spider-DK, Spider-Syn and Spider-Realistic dataset. Notably, SQL-R1 exhibits superior performance relative to above NL2SQL solutions, which further prove the generalization and robustness of a under different database and query requirements. Additional experimental results on other NL2SQL benchmarks (e.g., Spider2.0) can be found in Table 6 and Section B.

**Complexity Analysis.** We performed a statistical analysis of accuracy rates across different difficulty levels based on the BIRD-Dev dataset. The results in Table 3 demonstrate that SQL-R1 significantly outperforms the baseline across all three subsets: Simple, Moderate, and Challenging, indicating its robust capability in reasoning and generation for various SQL generation tasks. When we scaled SQL-R1 from 7B to 14B parameters, the accuracy rates for Simple and Moderate levels experienced minimal changes of just 0.3% and a decrease of 1.1%, respectively. However, a noteworthy improvement was observed in the Challenging category, where accuracy increased from 51.0% to 56.5%, marking a gain of 5.5 percentage points. This finding suggests that enhancing model capacity primarily addresses the generalization challenges associated with complex queries. Moreover, it aligns with existing research that indicates larger models are better at capturing long-range dependencies, particularly in scenarios involving multiple table joins, nested subqueries, or complex aggregations, highlighting the importance of SQL-R1's size in improving performance on sophisticated SQL generation tasks.

**Insights of Performance and Model Scale Trade-offs.** As illustrated in Figure 2, we investigated the relationship between performance and model size utilizing on BIRD development dataset. The comparative analysis encompasses various model types, including NL2SQL models, reasoning LLMs and general LLMs. For GPT-4, GPT-4o, GPT-4o-mini and Gemni-1.5-Pro, we refer to the parametric description of [22, 29, 41]. The findings demonstrate that when utilizing the 7B model as a foundational model, SQL-R1 attains accuracy levels that surpass those of larger-scale models, particularly. This underscores the efficacy of the proposed model in optimizing the performance of NL2SQL while ensuring that cost efficiency is maintained. Additionally, Section B.1 provides some detailed results of different base models.

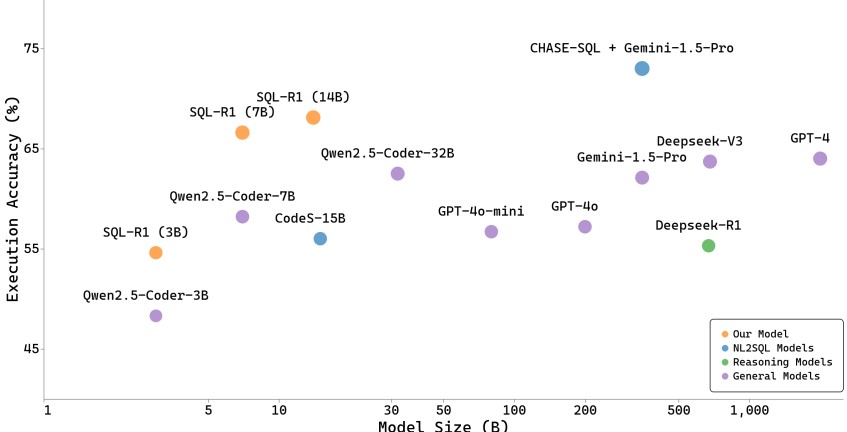

Figure 2: Performance and model scale on the BIRD-Dev dataset.

Table 4: Execution accuracy (%) of models with different cold start strategy. The *Reasoning Instruction* column is applied represents the instruction applied SFT process.

| Model | SFT Data | Reasoning Instruction | Spider (Dev) | Spider (Test) | BIRD (Dev) |
|---|---|---|---|---|---|
| Qwen2.5-Coder-7B | - | ✗ | 77.4 | 79.4 | 58.2 |
| Qwen2.5-Coder-7B | SynSQL-200K | ✓ | 82.7 | 83.3 | 57.0 |
| Qwen2.5-Coder-14B | - | ✗ | 87.0 | 88.0 | 66.1 |
| OmniSQL-7B [22] | SynSQL-2.5M | ✗ | 85.5 | 88.9 | 66.1 |
| OmniSQL-14B [22] | SynSQL-2.5M | ✗ | 86.2 | 88.3 | 65.9 |
| SQL-R1 + Qwen2.5-Coder-7B | - | ✗ | 84.5 | 86.1 | **63.1** |
| SQL-R1 + Qwen2.5-Coder-7B | SynSQL-200K | ✓ | 84.7 | 86.4 | 59.2 |
| SQL-R1 + Qwen2.5-Coder-14B | - | ✗ | 86.7 | 88.1 | **67.1** |
| SQL-R1 + OmniSQL-7B | SynSQL-2.5M | ✗ | 87.6 | 88.7 | **66.6** |
| SQL-R1 + OmniSQL-14B | SynSQL-2.5M | ✗ | 86.4 | 87.6 | **66.6** |

**Case Study.**  To explore the impact of RL training on the actual NL2SQL reasoning, we select some examples for analysis on the BIRD development dataset. As illustrated in Figure 3 to Figure 8, the model demonstrates enhanced reasoning capabilities following RL training. In the handling of more challenging samples, the model exhibits a discernible top-down cognitive strategy in the reasoning of generating SQL queries. This observation substantiates that reinforcement learning can further improve the reasoning ability of the model in NL2SQL tasks.

Above all, for *Q1*, SQL-R1 achieves superior performance on Spider and BIRD benchmarks, demonstrating the effectiveness of reinforcement learning in optimizing NL2SQL reasoning and outperforming models based on closed-source LLMs. This confirms the feasibility of designing an RL-based algorithm for NL2SQL tasks.

## 3.3   Analysis of SFT Cold Start

In this experiment, we conducted a comprehensive evaluation on different baselines: the base model without any post-training, the model utilizing the conventional format for SFT (e.g., OmniSQL-7B & OmniSQL-14B), and the model employing a reasoning instruction format for SFT.

Based on the findings demonstrated in Table 4, several implications can be discerned: Firstly, the quantity of synthesized data significantly influences the performance and generalization capabilities of SFT, as well as the effectiveness of subsequent RL training. For instance, the SQL-R1 7B model, which was initiated using SynSQL-2.5M, demonstrates superior performance compared to the model that trained with SynSQL-200K. Moreover, the provenance of the training data also plays a critical role in determining the performance of SFT. Notably, models that rely on synthetic data for SFT

Table 5: Ablation study of reward components on BIRD-Dev dataset.

| Reward Function | Accuracy (%) |
|---|---|
| Qwen2.5-Coder-7B | 58.2 |
| $S_f + S_e + S_r + S_l$ | **63.1** |
| - w/o $S_f$ (Format Score) | 60.4 ($\downarrow$ 2.7) |
| - w/o $S_e$ (Execution Score) | 60.7 ($\downarrow$ 2.4) |
| - w/o $S_r$ (Result Score) | 62.4 ($\downarrow$ 0.7) |
| - w/o $S_l$ (Length Score) | 61.0 ($\downarrow$ 2.1) |

Table 6: Execution accuracy (%) of different models on Spider2.0 SQLite subset.

| Model | Accuracy (%) |
|---|---|
| Qwen2.5-Coder-7B | 2.2 |
| GPT-4o | 15.6 |
| Deepseek-V3 | 15.6 |
| OmniSQL-7B | 10.4 |
| **SQL-R1-7B** | **20.0** |

exhibited only marginal gains when evaluated on the BIRD-Dev dataset. Specifically, the SFT methodology utilizing synthetic data did not yield an improvement in accuracy for the 7B model, even when the inference format adhered to the established instructions and utilized a training set of 200K instances. Additionally, the SFT training results for the 14B models in both Reasoning-SQL and OmniSQL underscore the importance of using a training set derived from a consistent source for SFT.

Thus, in response to **Q2**, these findings imply that SFT cold-start training is not universally essential for RL-based NL2SQL models. Its effectiveness is contingent upon the origin and volume of the training data. Future investigations should focus on delineating the optimal conditions that enable cold-start training to confer significant benefits, particularly with respect to data source and volume.

In addressing **Q3**, experimental results corroborate the assertion that synthetic data engineering plays a pivotal role in augmenting the reasoning capabilities of NL2SQL models, especially when utilized in conjunction with RL. The synergistic effect of synthetic data and RL training reveals marked enhancements in model performance. For instance, SQL-R1, when trained on SynSQL-Complex-5K, surpasses the performance of both the 7B and 14B baseline models. This finding elucidates the substantial potential inherent in synthetic data engineering to enhance model capabilities. Ongoing research endeavors will persist in exploring and refining strategies for the integration of synthetic data into the training regimen, with the aim of further advancing model performance and generalization.

### 3.4 Ablation Study of Reward Components

We conducted an ablation experiment on the BIRD development dataset to verify the proposed reinforcement learning reward mechanism's effectiveness. The experiment sequentially removed individual reward components from the comprehensive reward function while maintaining the parameter settings established in Section 3.1. As presented in Table 5, the result demonstrates that omitting any reward component from the original reward function adversely impacts reasoning performance. This underscores the critical importance of execution feedback and result reward in the model training process. Section B.3 and B.4 provide more details of the ablation study.

## 4 Related Works

**NL2SQL Methods.** Recent advancements in large language models have significantly propelled the development of NL2SQL translation techniques [1, 2, 42]. Current research focuses on optimizing various components of the NL2SQL workflow, including pre-processing modules like schema linking [5], translation strategies [43–49], and post-processing modules like SQL correction [8, 9, 12] and self-consistency for SQL selection [10, 16, 22]. However, existing NL2SQL models primarily rely on supervised fine-tuning, which may exhibit instability in domain adaptation and new scenario fitting, as they heavily depend on the knowledge of database schema and structure from the training data [13, 15, 50]. Additionally, the lack of interpretability in the reasoning logic of these NL2SQL models limits their application in high-risk scenarios or complex database requirements [51]. To address these limitations, we explore the application of reinforcement learning to enhance the reasoning and generalization capabilities of NL2SQL models. By dynamically adjusting the model's decision-making strategy through interaction with the environment, we aim to improve its performance in complex database scenarios and ensure better domain adaptation and interpretability.

**Reinforcement Learning for LLM Reasoning.** Recent research on reinforcement learning for LLMs has increasingly concentrated on enhancing reasoning capabilities and optimizing interaction with the external environment to improve proficiency in complex, multi-step reasoning tasks [52–54]. In recent months, DeepSeek-R1 [17] have illustrated the effectiveness of group relative policy optimization (GRPO), a novel and efficient reinforcement algorithm in cultivating robust reasoning behaviors in LLMs. By focusing exclusively on the correctness of the final answer, GRPO facilitates the internalization of intermediate reasoning steps without the necessity of explicit supervision, achieving notable success across various domains, including mathematics [55], finance [18], programming [56], user interface [57, 58] and vision tasks [59]. Concurrently, some studies have explored integrating external tools, such as retrieval [31] and search engines [19], into reinforcement learning frameworks for LLM reasoning, primarily focusing on training models to effectively employ a single, general-purpose tool to support specific reasoning chains. In light of these advancements, we aim to extend LLM reasoning capabilities to the NL2SQL task by adopting a reinforcement learning approach that dynamically adjusts the model's decision-making strategy through interaction with the environment, thereby enhancing its ability to manage complex semantics and improve generalization and domain adaptation capabilities.

## 5 Limitations

At present, this study still has the following limitations:

**Supported Database Dialect.** The current iteration of SQL-R1 has primarily been trained and evaluated on datasets that utilize the SQLite dialect. However, real-world databases often encompass various database dialects (e.g., Snowflake and DuckDB). Investigating and enhancing the generalization capabilities of NL2SQL models across these diverse database dialects presents a significant and valuable challenge for future research directions.

**Experiment.** We conduct experiments on Qwen2.5-Coder series models. However, due to the rapid advancements in this field, our evaluation is limited to the capabilities of these representative LLMs and we can not cover a broader range of new LLMs available currently (e.g., Llama4). We provide an initial test result of more models on Section B.1, and we will keep extending our work on many more base LLMs.

## 6 Conclusion

In this work, we propose SQL-R1, a novel NL2SQL reasoning model trained via reinforcement learning (RL), addressing key challenges in semantic understanding, reasoning, and generalization for complex database scenarios. By integrating dynamic reward mechanisms, cold start strategies, and sustainable data engineering, SQL-R1 achieves state-of-the-art performance on benchmark datasets (88.6% accuracy on Spider-Test and 67.1% on BIRD) while generating interpretable reasoning traces. Our study demonstrates the effectiveness of RL in enhancing model generalization and reducing domain adaptation costs, providing transparency for high-risk applications. Future work will focus on improving model interpretability, expanding multi-table joint capabilities, and exploring synthetic data generation to support scalable training. This research advances the practical usability of NL2SQL systems by bridging the gap between reasoning capability and real-world applicability.

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

# Appendix

## A Detailed Case Study

---

**Example for NL2SQL Reasoning - No RL Training - Category: Challenge**

**Question: Tally the toxicology element of the 4th atom of each molecule that was carcinogenic.**

Let's break down the question and the database schema step by step to generate the appropriate SQL query.

1. **Identify the Relevant Tables and Columns:**
- We need to identify the molecules that are carcinogenic (label = '+').
- We need to find the 4th atom of each molecule.
- We need to determine the element of the 4th atom.
- We need to tally the toxicology element.

2. **Filter Molecules by Carcinogenicity:**
- We will use the 'molecule' table to filter molecules where 'label = '+''.

3. **Find the 4th Atom of Each Molecule:**
- We will use the 'atom' table and the 'connected' table to find the 4th atom. The 4th atom's 'atom_id' will be the 7th character of the 'atom_id' column (since 'atom_id' follows the pattern 'TR000_1').

4. **Determine the Element of the 4th Atom:**
- We will use the 'atom' table to get the 'element' of the 4th atom.

5. **Tally the Toxicology Element:**
- We will use the 'element' column from the 'atom' table to tally the toxicology element.

6. **Construct the SQL Query:**
- We will join the 'molecule', 'atom', 'connected', and 'atom' tables to get the required information.

Here is the SQL query that addresses the question:
```sql
SELECT a.element FROM molecule m JOIN atom a ON m.molecule_id
= a.molecule_id JOIN connected c ON a.atom_id = c.atom_id
WHERE m.label = '+' AND a.atom_id LIKE 'TR000_4' GROUP BY
a.element; ✗
```
Explanation:
- 'SELECT a.element': Selects the element of the 4th atom.
- 'FROM molecule m': Joins with the 'molecule' table to filter molecules by their label.
- 'JOIN atom a ON m.molecule_id = a.molecule_id': Joins with the 'atom' table to get the atoms of each molecule.
- 'JOIN connected c ON a.atom_id = c.atom_id': Joins with the 'connected' table to find the connections between atoms.
- 'WHERE m.label = '+' AND a.atom_id LIKE 'TR000_4'': Filters the atoms to only include the 4th atom ('atom_id' starting with 'TR000_4').
- 'GROUP BY a.element': Groups the results by the element to tally the number of each element.

This query will provide the tally of the toxicology element of the 4th atom of each carcinogenic molecule.

---

Figure 3: Example for NL2SQL Reasoning - No RL Training - Challenge Sample

**Example for NL2SQL Reasoning - RL Training - Category: Challenge**

**Question: Tally the toxicology element of the 4th atom of each molecule that was carcinogenic.**

<think>

To answer the given question, we need to follow these steps:

1. **Identify the Molecules that are Carcinogenic**: According to the question, molecules that are carcinogenic have a label of '+' in the 'molecule' table.

2. **Identify the 4th Atom of Each Molecule**: The 4th atom is identified by the 'atom_id' where the substring of 'atom_id' starting from the 7th character and having a length of 1 is '4'.

3. **Retrieve the Element of the 4th Atom**: Once we have identified the 4th atom, we need to retrieve the element associated with it from the 'atom' table.

4. **Tally the Toxicology Element**: This step is not explicitly clear from the question, but it seems to imply that we need to count or list the elements of the 4th atoms of the carcinogenic molecules. Let's break down the steps into SQL queries:

### Step 1: Identify the Molecules that are Carcinogenic

```sql
SELECT molecule_id FROM molecule WHERE label = '+';
```

### Step 2: Identify the 4th Atom of Each Molecule

To find the 4th atom, we need to filter the 'atom_id' where the substring starting from the 7th character and having a length of 1 is '4'. We can use the 'SUBSTR' function for this.

```sql
SELECT atom_id FROM atom WHERE SUBSTR(atom_id, 7, 1) = '4';
```

### Step 3: Retrieve the Element of the 4th Atom

We need to join the 'atom' table with the 'connected' table to find the 4th atom and then retrieve its element.

```sql
SELECT atom.element FROM atom JOIN connected ON atom.atom_id
= connected.atom_id WHERE SUBSTR(atom.atom_id, 7, 1) = '4';
```

### Step 4: Tally the Toxicology Element. To tally the toxicology element, we need to count or list the unique elements of the 4th atoms. We can use the 'GROUP BY' clause to achieve this.

```sql
SELECT atom.element FROM atom JOIN connected ON atom.atom_id
= connected.atom_id WHERE SUBSTR(atom.atom_id, 7, 1) = '4'
GROUP BY atom.element;
```

### Combining All Steps: We need to combine all the steps into a single query that filters the carcinogenic molecules, identifies the 4th atom, retrieves its element, and tallies the unique elements.

```sql
SELECT atom.element FROM atom JOIN connected ON atom.atom_id
= connected.atom_id JOIN molecule ON atom.molecule_id =
molecule.molecule_id WHERE SUBSTR(atom.atom_id, 7, 1) = '4'
AND molecule.label = '+' GROUP BY atom.element;
```

</think>

<answer>

```sql
SELECT atom.element FROM atom JOIN connected ON atom.atom_id
= connected.atom_id JOIN molecule ON atom.molecule_id =
molecule.molecule_id WHERE SUBSTR(atom.atom_id, 7, 1) = '4'
AND molecule.label = '+' GROUP BY atom.element;✓
```

</answer>

Figure 4: Example for NL2SQL Reasoning - RL Training - Challenge Sample

**Example for NL2SQL Reasoning - No RL Training - Category: Moderate**

**Question: What is the display name of the user who last edited the post "Examples for teaching: Correlation does not mean causation"**

To determine the display name of the user who last edited the post with the title Ëxamples for teaching: Correlation does not mean causation,̈ we need to follow these steps:

1. **Identify the relevant table**: The information about posts, including their titles and the user who last edited them, is stored in the 'posts' table. 2. **Locate the correct post by title**: We need to filter the posts to find the one with the title Ëxamples for teaching: Correlation does not mean causation.̈

3. **Retrieve the last editor's display name**: Once we have identified the post, we can access the 'LastEditorDisplayName' column to get the display name of the user who last edited the post.

Here is the SQL query that accomplishes this:

```sql
SELECT LastEditorDisplayName FROM posts WHERE Title =
'Examples for teaching:  Correlation does not mean causation';
✗
```

This query selects the 'LastEditorDisplayName' from the 'posts' table where the 'Title' matches the specified title.

Figure 5: Example for NL2SQL Reasoning - No RL Training - Moderate Sample

**Example for NL2SQL Reasoning - RL Training - Category: Moderate**

**Question: What is the display name of the user who last edited the post "Examples for teaching: Correlation does not mean causation"**

<think>

To answer this question, we need to find the display name of the user who last edited a specific post. Here are the steps to derive the SQL query:

1. **Identify the Post**: We start by identifying the post with the title Ëxamples for teaching: Correlation does not mean causation.. This information is available in the 'posts' table under the 'Title' column.

2. **Find the Last Editor**: The question asks for the user who last edited the post, which is referred to by 'LastEditorUserId' in the 'posts' table. This 'LastEditorUserId' is a foreign key referencing the 'Id' column in the 'users' table.

3. **Get the Display Name**: Once we have the 'LastEditorUserId', we need to find the corresponding display name of the user. This information is available in the 'users' table under the 'DisplayName' column.

4. **Formulate the Query**:
- **Select the Required Column**: We need to select the 'DisplayName' from the 'users' table.
- **Join the Tables**: We need to join the 'posts' table (aliased as 'T1') with the 'users' table (aliased as 'T2') to link the post to the user who last edited it. This is done using the 'LastEditorUserId' in the 'posts' table and the 'Id' in the 'users' table.
- **Filter the Post**: We need to filter the posts to find the one with the title "Examples for teaching: Correlation does not mean causation".

5. **Construct the SQL Query**:
- **SELECT**: We select the 'DisplayName' from the 'users' table.
- **FROM**: We start from the 'posts' table (aliased as 'T1').
- **INNER JOIN**: We join the 'users' table (aliased as 'T2') on the condition that 'T1.LastEditorUserId' equals 'T2.Id'.
- **WHERE**: We filter the results to find the post with the title "Examples for teaching: Correlation does not mean causation".

Accordingly, the final SQL query is:

```sql
SELECT T2.DisplayName FROM posts AS T1 INNER JOIN users AS T2
ON T1.LastEditorUserId = T2.Id WHERE T1.Title = 'Examples for
teaching:  Correlation does not mean causation';
```

</think>

<answer>

```sql
SELECT T2.DisplayName FROM posts AS T1 INNER JOIN users AS T2
ON T1.LastEditorUserId = T2.Id WHERE T1.Title = 'Examples for
teaching:  Correlation does not mean causation';✓
```

</answer>

Figure 6: Example for NL2SQL Reasoning - No RL Training - Moderate Sample

**Question: How many female patients born in 1964 were admitted to the hospital? List them by ID.**

To answer this question, we need to generate a SQL query that will filter the patients based on the given conditions and then count the number of patients who meet these conditions. Here are the steps to derive the SQL query:

1. **Identify the Relevant Table**: The information about patients, including their sex, birth year, and hospital admission status, is stored in the 'Patient' table.

2. **Filter Conditions**:
- The patient must be female, which is represented by 'SEX = 'F''.
- The patient must be born in 1964, which can be determined using the 'Birthday' column and the 'strftime' function to extract the year.
- The patient must have been admitted to the hospital, which is represented by 'Admission = '+''.

3. **Count the Number of Patients**: Since the question asks for the number of patients, we will use the 'COUNT' function to count the number of patient IDs that meet the specified conditions.

4. **Select the Patient IDs**: The question also asks to list the patients by ID, so we will include the 'ID' column in the 'SELECT' clause.

Here is the SQL query that satisfies all these requirements:

```sql
SELECT COUNT(ID) AS NumberOfPatients, ID FROM Patient WHERE
SEX = 'F' AND strftime('%Y', Birthday) = '1964' AND Admission
= '+'; ✗
```

This query will return the number of female patients born in 1964 who were admitted to the hospital and list their IDs

Figure 7: Example for NL2SQL Reasoning - No RL Training - Simple Sample

> **Example for NL2SQL Reasoning - RL Training - Category: Simple**
>
> **Question: How many female patients born in 1964 were admitted to the hospital? List them by ID.**
>
> \<think\>
>
> To generate the SQL query for the given question, we need to follow these steps:
>
> 1. **Identify the Relevant Table**: The question is about patients, so we need to focus on the 'Patient' table.
>
> 2. **Determine the Conditions**:
>
> - "female" refers to 'SEX = 'F''.
>
> - "born in 1964" refers to 'YEAR(Birthday) = 1964'. However, SQLite does not have a built-in 'YEAR' function, so we need to use the 'strftime' function to extract the year from the 'Birthday' column.
>
> - "admitted to the hospital" refers to 'Admission = '+''.
>
> 3. **Select the Required Information**: The question asks for the IDs of the patients who meet the specified conditions. Therefore, we need to select the 'ID' column from the 'Patient' table.
>
> 4. **Formulate the SQL Query**:
>
> - Use the 'SELECT' statement to specify that we want to retrieve the 'ID' column.
>
> - Use the 'FROM' clause to specify the 'Patient' table.
>
> - Use the 'WHERE' clause to apply the conditions:
>
> - 'SEX = 'F'' for female patients.
>
> - 'strftime('%Y', Birthday) = '1964'' to filter patients born in 1964.
>
> - 'Admission = '+'' to filter patients who were admitted to the hospital.
>
> 5. **Construct the Final Query**:
>
> - Start with the 'SELECT' statement to choose the 'ID' column.
>
> - Use the 'FROM' clause to specify the 'Patient' table.
>
> - Use the 'WHERE' clause to include the conditions: 'SEX = 'F'', 'strftime('%Y', Birthday) = '1964'', and 'Admission = '+''.
>
> Accordingly, the final SQL query is:
>
> ```sql
> SELECT ID FROM Patient WHERE SEX = 'F' AND strftime('%Y',
> Birthday) = '1964' AND Admission = '+';
> ```
>
> \</think\>
>
> \<answer\>
>
> ```sql
> SELECT ID FROM Patient WHERE SEX = 'F' AND strftime('%Y',
> Birthday) = '1964' AND Admission = '+'; ✓
> ```
>
> \</answer\>

Figure 8: Example for NL2SQL Reasoning - RL Training - Simple Sample

# B Additional Results and Analysis

## B.1 Analysis and Comparison of Base LLMs

Table 7: Execution accuracy (%) of different base LLMs on Spider and BIRD benchmark.

| Base Model | Spider (Dev) | Spider (Test) | BIRD (Dev) |
|---|---|---|---|
| Qwen2.5-Coder-3B | 77.0 | 77.2 | 50.5 |
| Qwen2.5-Coder-7B | 77.4 | 79.4 | 58.2 |
| Qwen2.5-Coder-14B | 87.0 | 88.0 | 66.1 |
| Qwen3-8B (Thinking Mode) | - | - | 50.8 |
| Qwen3-14B (Thinking Mode) | - | - | 51.8 |
| **SQL-R1 + Qwen2.5-Coder-3B** | **78.1** | **78.9** | **54.6** |
| **SQL-R1 + Qwen2.5-Coder-7B** | **87.6** | **88.7** | **63.1** |
| **SQL-R1 + Qwen2.5-Coder-14B** | **86.7** | **88.1** | **67.1** |

As shown in Table 7, we found that smaller models benefit significantly more from RL training than their larger counterparts. Specifically, SQL-R1 trained on Qwen2.5-Coder-3B and Qwen2.5-Coder-7B demonstrated substantial accuracy improvements on both the Spider and BIRD benchmarks. In contrast, the Qwen2.5-Coder-14B model exhibited a relatively minor improvement. These results suggest that RL is particularly effective for smaller models, likely due to their greater capacity for performance enhancement through targeted training. This observation leads to two critical insights: first, the potential for deploying RL on smaller base models could yield significant performance gains with higher cost efficiency, enhancing their practicality for real-world applications; second, larger-scale models, such as the 14B variant, may necessitate more extensive training data and advanced training strategies to realize meaningful improvements. Future research could thus focus on optimizing RL methodologies for smaller models to maximize their potential, while also exploring sophisticated training techniques and larger datasets to leverage the capabilities of more complex LLMs fully.

## B.2 Analysis and Comparison of Efficiency

We assessed the latency of SQL-R1 on the BIRD-Dev dataset, maintaining the same inference settings and evaluation metric as outlined in Section 3.1.

Table 8: Efficiency comparison of different NL2SQL methods on BIRD-dev dataset. The base model of XiYan-SQL and SQL-R1 is Qwen2.5-Coder-7B.

| NL2SQL Method | Candidate Selection | Latency (s) / Question | Total Tokens (K) Question | Accuracy (%) |
|---|---|---|---|---|
| Qwen2.5-Coder-7B | Greedy Search | 0.3 | 2.5 | 58.2 |
| XiYan-SQL | Greedy Search | 0.5 | 4.1 | 62.1 |
| CHESS | Greedy Search | 251.3 | 320.8 | 61.5 |
| **SQL-R1 (Ours)** | **Greedy Search** | **0.4** | **3.1** | **63.7** |
| **SQL-R1 (Ours)** | **Self-Consistency** | **1.1** | **23.1** | **66.6** |

As demonstrated in Table 8, the self-consistency method employing 8 candidates adds only an average of 0.7 seconds to each query than greedy search method, while improving execution accuracy by 2.9%, which represents a favorable trade-off. Additionally, We compared SQL-R1 with XiYan-SQL [60] using the same base model and related baselines, and observed that SQL-R1 achieved higher accuracy under the same conditions. In contrast to CHESS [7], which has an execution accuracy of 61.5% and employs a multi-component workflow, the end-to-end SQL-R1 model achieves an accuracy of 66.6%. SQL-R1 not only shows superior execution accuracy but also offers significant advantages in terms of latency and token usage efficiency.

## B.3 Ablation Study of SQL Candidate Rollouts in Reasoning

To assess the efficacy of the self-consistency in the inference, we conducted ablation studies on the Qwen2.5-Coder-3B and Qwen2.5-Coder-7B base models, aiming to determine the optimal number of SQL candidate rollouts generated under varied temperature settings designed to enhance diversity. The results illustrated in Figure 9 reveal that setting the number of SQL candidates to 8 strikes the ideal balance between diversity and computational efficiency. In the case of the 3B model, configurations with fewer than eight candidates resulted in diminished execution accuracy due to insufficient diversity, whereas exceeding 8 candidates led to a plateau in accuracy alongside increased computational overhead. The 7B model demonstrated relatively stable accuracy across different candidate counts but achieved its highest accuracy at the same optimal setting. These findings substantiate that the parameter configuration of 8 SQL candidates and a temperature setting of 0.8 effectively enhances SQL-R1's performance, successfully navigating the trade-off between candidate diversity and computational cost, thereby reinforcing the model's robustness in SQL query generation.

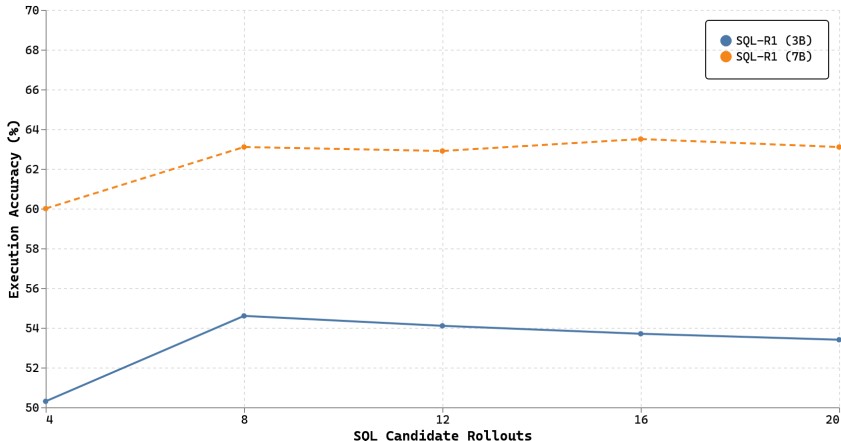

Figure 9: Performance of different scale of SQL candidate rollouts on BIRD-Dev dataset.

## B.4 Ablation Study of Database Value Retrieval for RL Training

Table 9: Ablation study of database value retrieval for RL training on BIRD-Dev dataset.

| Base Model | Use Database Values | Accuracy (%) |
|---|---|---|
| Qwen2.5-Coder-7B | - | 58.2 |
| SQL-R1 + Qwen2.5-Coder-7B | ✗ | 61.9 |
| **SQL-R1 + Qwen2.5-Coder-7B** | ✓ | **63.1** |

We conducted an ablation study to evaluate the effects of integrating database values during the RL training. As illustrated in the Table 9 , incorporating database values did not yield a statistically significant improvement in the model's overall performance. However, this integration resulted in an increased input sequence length, which consequently diminished training efficiency. These findings substantiate our decision to forgo the inclusion of representative value annotations during the training phase. This strategic choice enables the model to concentrate more effectively on schema linking, thereby enhancing its exploratory capabilities during the RL phase and ultimately leading to improved reasoning and generalization skills.

## B.5 Ablation Study of Reward Sensitivity

We conducted experiments on the BIRD-Dev dataset by adjusting the weights of various reward components to assess their impact on the model's performance.

The sensitivity analysis highlights crucial insights into reward function design for NL2SQL tasks. Minimal weights for Execution and Result rewards lead to performance degradation due to potential

Table 10: Ablation study of different setting of reward components.

| Reward Type | Set to 1.0 | Increase by 50% |
|---|---|---|
| $S_e$ (Execution Score) | 60.0 | - |
| $S_r$ (Result Score) | 61.3 | - |
| $S_l$ (Length Score) | 59.7 | 60.6 |

reward hacking, as the model exploits simpler patterns instead of developing robust reasoning. Conversely, excessively high reward weights during early training can destabilize learning by resulting in extreme rewards before solid reasoning is established. The model shows lower sensitivity to Result reward adjustments, indicating challenges in bridging semantic gaps despite successful exploration of SQL queries. Additionally, a fixed Length reward is ineffective for fostering deep reasoning, while our smooth sensitivity design maintains differentiation without causing training oscillations; however, excessive Length rewards may lead to verbose solutions. Overall, these findings reinforce the need for balanced reward calibration to promote stable learning and minimize exploitation in reinforcement learning.

## B.6 Visualization of Training Process

Figure 10 illustrates the progression of reward and response length throughout the training process on Qwen2.5-Coder-3B, Qwen2.5-Coder-7B and Qwen2.5-Coder-14B models.

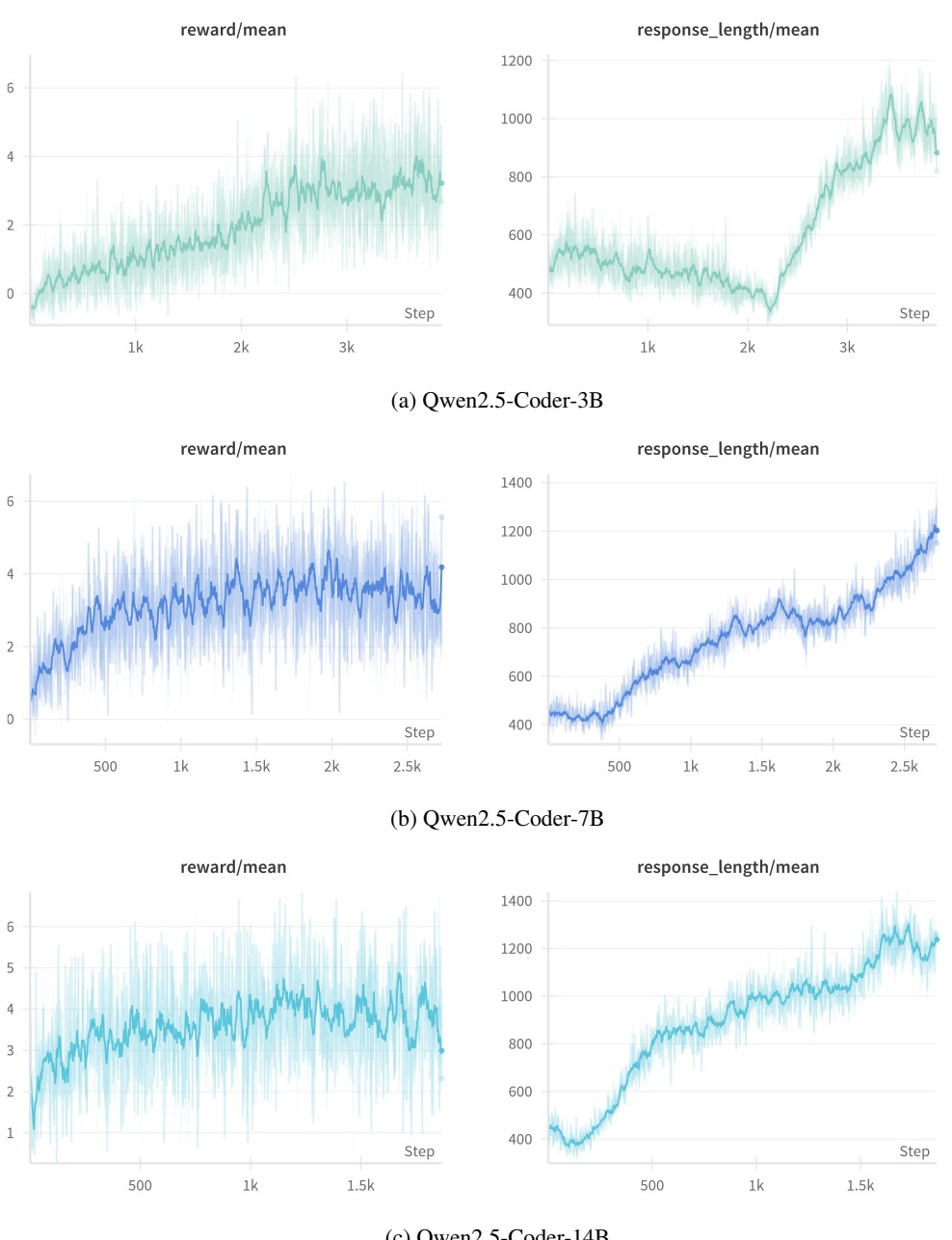

(a) Qwen2.5-Coder-3B

(b) Qwen2.5-Coder-7B

(c) Qwen2.5-Coder-14B

Figure 10: Visualization of training process on different scale base models.

## C   Prompt Templates

**Prompt Template for RL Training**

You are a helpful SQL expert assistant.

The assistant first thinks about how to write the SQL query by analyzing the question, database schema and external knowledge, then provides the final SQL query. The reasoning process and SQL query are enclosed within <think> </think> and <answer> </answer> tags respectively. The answer must contain the SQL query within ```sql...``` tags.

Database Schema: {schema}

External Knowledge: {external_knowledge}

For example:

<think>

To translate the given natural language question into an executable SQLite query, we need to follow these detailed steps:

1.  **Identify Key Elements**: The question queries for code snippets that are both complicated (complexity score > 5) and public ('is_public' = 1). We need to retrieve their descriptions and complexity scores.

2. **Focus on Relevant Tables**: The 'code_snippets' table contains the necessary fields ('description', 'complexity', 'is_public').

3.  **Construct the Query**: We should select the required fields ('description' and 'complexity') from the 'code_snippets' table. We also apply the conditions specified in the question to filter the results.

4. **Ordering**: The reference solution includes an 'ORDER BY' clause to sort results by complexity in descending order, which is a reasonable way to present the data to highlight the most complex snippets first.

5. **Final Query Construction**: Putting all this together into a SQL query.

</think>

<answer>

Here's how the query can be written:

```sql
SELECT description, complexity FROM code_snippets WHERE
complexity > 5 AND is_public = 1 ORDER BY complexity DESC;
```

This query retrieves the descriptions and complexity scores of code snippets that are both complicated (complexity > 5) and publicly available ('is_public' = 1), sorted by complexity in descending order.

This solution is straightforward and precisely matches the requirements of the question. It avoids unnecessary complexities, such as joining or selecting columns not relevant to the query itself.

</answer>

Question: {question}

Figure 11: Prompt Template for RL Training

Figure 12: Prompt Template for SFT Training

