# OpenReview forum: "SQL-R1: Training Natural Language to SQL Reasoning Model By Reinforcement Learning"
_NeurIPS.cc/2025/Conference — NeurIPS 2025 poster_

### Official Review · Reviewer_YmQu · 2025-06-13

**Clarity:** 3
**Significance:** 3
**Originality:** 3
**Rating:** 4
**Confidence:** 3

**Summary:**

This paper introduces SQL-R1, a novel approach for Natural Language to SQL (NL2SQL) that leverages reinforcement learning (RL) fine-tuning of large language models (LLMs). Motivated by the success of RL-based training in improving reasoning capabilities of LLMs, the authors employ Group Relative Policy Optimization (GRPO) to enhance SQL generation performance. The work also addresses several key research questions such as the optimal cold start strategy for RL fine-tuning in the NL2SQL domain, and the characteristics and requirements of high-quality training data for learning effective NL2SQL models. The authors demonstrate that SQL-R1 achieves state-of-the-art performance on two standard benchmarks, Spider and BIRD.

**Questions:**

**Missing Competitive Baselines in Figure 2.** Figure 2 appears to omit several recent competitive methods that could significantly impact the performance comparison. Notable missing baselines include SQL-o1 and Alpha-SQL, among others. This omission may present an incomplete picture of the current state-of-the-art and could mislead readers about the relative performance of SQL-R1. Could the authors include these methods in their comparison or provide justification for their exclusion?

**Selective Dataset Analysis.** In lines 250-251, the authors discuss the effectiveness of selective datasets for RL fine-tuning by referencing base model results which are not in the main paper. Can you add the guidance where the reader check for the base model results? Additionally, to show the effectiveness of the selective dataset, it would be better to compare the SQL-R1 trained from the selective dataset and default dataset. Can you test this also?

**Performance Analysis by SQL Complexity.** Given that RL fine-tuning with specialized rewards should theoretically be more beneficial for complex reasoning tasks, it would be valuable to see a breakdown of performance across different SQL complexity levels (Easy/Medium/Hard). This analysis would help readers understand when and why the proposed method provides the most benefit. Can you provide these level-wise performance comparisons?

**Relationship to Recent RL Fine-tuning Effectiveness Debates.** Recent work [1] has raised important questions about the effectiveness of RL fine-tuning, demonstrating that reasoning-optimized models can underperform on simple tasks while showing improvements on medium-complexity problems when given additional computational budget. Given these findings, how does SQL-R1's performance align with this complexity-dependent effectiveness pattern? Do the authors observe similar patterns in SQL-R1's results? More broadly, can the authors provide analysis and discussion of their method through this lens, including whether SQL-R1 exhibits the complexity-dependent benefits and limitations observed in recent RL fine-tuning research?

[1] https://machinelearning.apple.com/research/illusion-of-thinking

**Ethical Concerns:**

["NO or VERY MINOR ethics concerns only"]

**Final Justification:**

My main concern was the performance improvement pattern through RL fine-tuning such as higher improvement on the complex patterns while decreasing the performance on the simpler one. However, in their rebuttal, I can find the performance improvements are generally found for diverse difficulties of the queries. Hence, I raised my score from borderline reject to borderline acceptance.

**Limitations:**

yes

**Paper Formatting Concerns:**

I did not notice any major formatting issues in this paper.

**Quality:**

3

**Strengths And Weaknesses:**

## Strengths

**Timely Application of RL Fine-tuning to NL2SQL.** The application of reinforcement learning fine-tuning to NL2SQL is well-motivated given the recent success of RL methods in improving LLM reasoning capabilities. Since NL2SQL requires complex reasoning to generate accurate SQL statements, this approach represents a natural and promising direction for the field.

**Comprehensive Investigation of Key Research Questions.** Beyond proposing the core method, the authors systematically investigate important practical questions, including the necessity of supervised fine-tuning as a prerequisite (cold start problem) and the requirement of high-quality training data. These investigations provide valuable insights that advance understanding of RL fine-tuning for structured generation tasks.

**Thorough Experimental Evaluation.** The paper includes comprehensive comparisons against diverse NL2SQL baselines across multiple datasets and model scales. The ablation studies on reward design components are particularly valuable for understanding which aspects of the approach contribute most to performance gains.


## Weaknesses

**Limited Performance Gains.** While the method achieves competitive results, the improvements over existing state-of-the-art methods are modest and inconsistent. For instance, on Spider test set with Qwen2.5-Coder-7B, SQL-R1 achieves 88.7% versus OmniSQL's 88.9%. The performance gaps vary across different experimental settings, raising questions about the practical significance of the approach.

**Insufficient Baseline Analysis.** The experimental section compares against numerous prior methods, but the paper lacks adequate explanation of these baselines in the related work or methodology sections. Readers would benefit from understanding the key differences between SQL-R1 and these comparative methods, particularly how they differ in their approach to the NL2SQL problem.

---

> ### Author Rebuttal · Authors · 2025-07-31
>
> Dear Reviewer,
>
> We sincerely thank you for your thorough review and valuable feedback. Your insightful comments will help us significantly improve the quality of our paper.
>
> **1. For Missing Competitive Baselines in Figure 2 (In response to Q1)**
>
> * **Our Response**: We sincerely thank you for this valuable suggestion. In Figure 2, we mainly highlight the advantages of SQL-R1 over the NL2SQL method using different scale-based models, so all the compared models are not fully presented. In Table 1, we highlight the above baselines (including Alpha-SQL and SQL-o1) and their related information, and compare them. We really appreciate your valuable comments and will modify and supplement them in the revised version.
>
> **2. For Selective Dataset Analysis (In response to Q2)**
>
> * **Our Response**: We appreciate your question about this place.  We must clarify that the models of all scales of SQL-R1 use the SynSQL-Complex-5K dataset for the RL phase. Therefore, all the results are dependent on this dataset.  Also, we strongly agree with your issues regarding RL training on the default dataset.  However, the original dataset contains 2.5M pieces of training data, which would be a very expensive and time-consuming challenge for RL training (more than 2 months). We regret not being able to complete the above training and provide corresponding results now.
>
> * Additionally, we note that the proposed work aims to use a small amount of synthetic data to stimulate the model's NL2SQL deep reasoning ability. The data cost is also one requirement for the work's efficient operation. We will fully consider your comments and continue improving this area's work thereafter.
>
> **3. For the Performance Analysis by SQL Complexity (In response to Q3)**
>
> * **Our Response**: We appreciate your question about this place. We conducted a statistical analysis of the accuracy rates for various difficulty levels based on the BIRD development dataset, as presented in the table below:
>
> | Method | Simple | Moderate | Challenging | All |
> |---|---|---|---|---|
> | SQL-R1 (7B) | 72.1 | 60.8 | 51.0 | 66.6 |
> | SQL-R1 (14B) | 72.4 | 59.7 | 56.5 | 67.1 |
> | SuperSQL | 66.9 | 46.5 | 43.8 | 58.5 |
> | DAIL-SQL | 63.0 | 45.6 | 43.1 | 55.9 |
>
> * The results displayed show that SQL-R1 maintains a notable accuracy advantage over the current baseline across the three subsets: Simple, Moderate, and Challenging. This observation indicates that SQL-R1 exhibits a universally beneficial capability in reasoning and generation across different levels of SQL generation tasks.
>
> * Moreover, when we expanded SQL-R1 from a parameter size of 7B to 14B, the EX rates for the Simple and Moderate difficulty levels experienced a marginal change of only 0.3% and a decrease of 1.1%, respectively. However, it is essential to highlight the significant improvement in the Challenging difficulty category, where the accuracy surged from 51.0% to 56.5%, with an increase of 5.5 percentage points. This finding suggests that augmenting model capacity primarily alleviates the generalization bottleneck associated with high-difficulty queries.
>
> * This phenomenon corroborates previous research findings, which assert that larger-scale models are more adept at capturing long-range dependencies, particularly when queries involve multiple table joins, nested subqueries, or complex aggregations. The above insights underscore the critical role of model size of SQL-R1 in enhancing performance on intricate SQL generation tasks.
>
>
> **4. For the Relationship to Recent RL Fine-tuning Effectiveness Debates (In response to Q4)**
>
> * **Our Response**: We thank the reviewers for pointing out the insightful connection between SQL-R1 and the recent study on complexity-dependent RL effectiveness.  Below we provide some discussions:
>
> * **Complexity-Dependent Pattern Analysis**: We appreciate this insightful connection to recent research on RL fine-tuning effectiveness. While SQL-R1 shows some similarities in complexity-dependent patterns, our analysis reveals certain unique characteristics that differ from previous findings:
>
> | Query Complexity | Performance Pattern | Explanation |
> |-----------------|---------------------|-------------|
> | Simple Queries | Moderate Gains | RL provides limited benefits for straightforward SELECT-FROM-WHERE queries where base model capabilities are already strong |
> | Medium Complexity | Limited Improvement | RL shows modest gains on queries with joins and aggregations, as training data biases model toward more complex solutions |
> | Challenge Complexity | Significant Gains | RL training data's preference for complex samples leads model to excel at intricate nested queries, though risks overcomplicating simpler cases |
>
> * **Computational Budget Impact**: Based on our experimental results, we observe that increasing model capacity from 7B to 14B yields the highest relative gains (5.5% improvement) on challenging-complexity queries, while showing minimal impact on simple and moderate queries. This suggests that expanding computational resources primarily benefits the model's ability to handle more complex SQL generation tasks involving intricate joins, nested queries and aggregations.
>
> * In summary, SQL-R1 exhibits performance that is partially consistent with the above studies. For NL2SQL tasks, RL training can effectively improve the model's ability to solve higher complexity problems and trigger deeper thinking and reasoning. However, overstrengthening the ability to think about complex problems also leads to the risk of overthinking the model on simple problems. Balancing the above preferences during training will help the model to make more accurate judgments and draw inferences.
>
> **5. For the issue of Insufficient Baseline Analysis (In response to W2)**
>
> * **Our Response**: We sincerely appreciate your insightful question regarding baseline analysis. We are grateful for the opportunity to provide a comprehensive response to this important point.
>
> * **Comparsion with workflow-based NL2SQL methods**: In contrast to various multi-module workflow paradigms such as DIN-SQL, DAIL-SQL, MAC-SQL, SuperSQL, MCTS-SQL, OpenSearch-SQL and CHASE-SQL, SQL-R1 introduces a streamlined end-to-end decoding strategy centered on self-consistency, which mitigates the necessity for explicit task decomposition, complex schema linking, the generation of intermediate representations, debug processes, and Monte-Carlo tree searches.   Traditional pipeline-oriented systems predominantly depend on proprietary LLMs like GPT-4/4o and Gemini-1.5-Pro, which typically necessitate multiple iterations for refining SQL skeletons and often entail using external tools for schema retrieval and execution feedback. This reliance results in substantial computational overhead and heightened engineering complexity. Conversely, by leveraging a single open-source backbone, SQL-R1 achieves competitive accuracy metrics of 87.6% and 88.7% on the Spider development and test sets, respectively, and 66.6% on the BIRD development set. This approach simplifies deployment and diminishes token costs while upholding a robust performance profile.
>
> * **Comparsion with SFT-based NL2SQL methods**: SQL-R1 marks a significant advancement beyond traditional reliance on SFT methods, such as CodeS, DTS-SQL, CHESS, and OmniSQL. While these SFT-based approaches demonstrate strong memorization of common SQL patterns—reflected in their Spider test scores ranging from 79.4% to 88.9%—they often struggle with generalization on more challenging benchmarks like BIRD, where scores range from 54.6% to 67.1%. To overcome this limitation, SQL-R1 integrates self-consistency reranking during decoding, effectively bridging the performance gap between in-domain and cross-domain tasks. We will add these detailed explanations to the revised manuscript.
>
> We are deeply grateful to the reviewer for the constructive feedback. We have carefully addressed each concern with detailed responses and additional analyses. We hope these clarifications adequately address your questions and will incorporate all suggestions in our revised manuscript. We greatly appreciate your help in improving this work.
>
>
>
>
> Sincerely,
>
> The Authors

---

> > ### Comment · Reviewer_YmQu · 2025-08-02
> > **Reply to the rebuttal**
> >
> > I thank the authors for their detailed feedback. They addressed my concerns properly. Therefore, I raised my score.

---

> ### Author Response · Authors · 2025-08-05
>
> Dear Reviewer YmQu,
>
> We sincerely appreciate you for the raised score and the helpful review that enabled us to enhance the discussion of our work and strengthen our experimental analysis.
>
> Sincerely,
>
> The Authors

---

> > ### Comment · Reviewer_YmQu · 2025-08-05
> >
> > I share my more detailed comment for the authors rebuttal.
> >
> > My main concern was the performance improvement pattern through RL fine-tuning such as higher improvement on the complex patterns while decreasing the performance on the simpler one. However, in their rebuttal, I can find the performance improvements are generally found for diverse difficulties of the queries. Hence, I raised my score.

---

> > > ### Author Response · Authors · 2025-08-07
> > >
> > > Dear Reviewer YmQu,
> > >
> > > We sincerely appreciate your incisive feedback and the time you invested in scrutinizing every detail.  Your rigorous suggestions have pushed us to sharpen our exposition, solidify our experimental evidence, and confront the limits of our claims head-on. Thank you for your valuable contribution.
> > >
> > > Sincerely,
> > >
> > > The Authors

---

### Official Review · Reviewer_Bp1S · 2025-06-28

**Clarity:** 4
**Significance:** 2
**Originality:** 1
**Rating:** 2
**Confidence:** 5

**Summary:**

Inspired by the success of the RL training approach on language models, SQL-R1 uses GRPO to train open-source language models (Qwen-2.5 family) on the Text-to-SQL task. They introduce four rewards for the training, including format, executability, execution accuracy, and length reward, that, in combination, guide the RL training process. This work shows that an open-source model trained with RL can outperform the SFT-trained counterpart, reaching a comparable performance to some larger proprietary models.

**Questions:**

Q1) Figure 9 shows the performance of the 7B model reaching around 63% using self-consistency on the BIRD dev set for 8 SQL candidates, whereas Table 1 reports this as 66.6%. Could the authors clarify this discrepancy?

Q2) The results reported in Reasoning-SQL represent the zero-shot performance of the model without parallel sampling and self-consistency. However, Table 1 indicates the use of self-consistency, differing from the original report. Did the authors reproduce the results from Reasoning-SQL and other comparable works? It would be helpful to add a column reporting the sampling budget of each method for a clearer comparison.

Q3) How were the weights for the partial rewards determined? Specifically, considering the execution score computation depends on the format reward, how was the training conducted for the "w/o Sf" scenario in Table 3? Surprisingly, the Sr reward, which directly measures performance on the downstream task, seems to have the least impact. Could you elaborate on why this might be the case?

Q4) Moreover, I am interested in understanding the impact of the length reward on model performance. How does this reward specifically guide the model toward generating better SQL queries? Could you provide qualitative examples comparing the reasoning processes of models trained with and without this reward?

**Ethical Concerns:**

["NO or VERY MINOR ethics concerns only"]

**Final Justification:**

I thank the authors for the effort they put into responding to my questions. I have carefully read their responses to both my comments and those of the other reviewers.

Below, I outline several concerns that remain unresolved:

- Since the primary reward used in the RL training for SQL-R1 is *execution accuracy*, which is also the standard downstream metric for Text-to-SQL, the method does not introduce a novel task-specific adaptation for reinforcement learning in this context.

- The OmniSQL work has already demonstrated that training on its synthetic data improves the performance of open-source models on the Text-to-SQL task. Replacing supervised fine-tuning (SFT) with reinforcement learning (RL) on this data, without additional innovation, does not represent a significant contribution in my view.

Because of the two reasons above, the novelty of this work remains limited to me, specifically for a prestigious venue such as NeurIPS.

- In line with concern W2: because baseline comparisons can be highly sensitive to implementation details, evaluating the method on BIRD’s held-out test set remains essential for a fair and direct comparison across approaches.

- Despite the follow-up discussion, the concerns related to Q4 and W2 remain insufficiently addressed.

For these reasons, I would keep my original score.

**Limitations:**

The main limitation of the work is its lack of novelty, and I would appreciate it if the authors could elaborate on that.

**Paper Formatting Concerns:**

No major formatting concerns

**Quality:**

3

**Strengths And Weaknesses:**

Strengths:

S1) The paper is well-presented, easy to follow, and provides clear explanations of methodologies and findings.

S2) The experimental details are clearly explained and well-documented, helping with the reproducibility of the results.

S3) The authors have shown extensive experimental results across multiple model sizes, using two well-established and rigorous benchmarks for Text-to-SQL (Spider and BIRD)

S4) Ablation studies are presented for the training data, model size, and inference-budget, providing insights into the individual impact of each component on the overall performance.

Weaknesses:

W1) Novelty: The primary issue with this work is the lack of novelty. As cited by the paper itself, Reasoning-SQL has a very similar experimental setting and has already validated most of the claims made in this work. Specifically, Reasoning-SQL previously adopted GRPO for Text-to-SQL and employed rewards very similar to the format, executability, and execution accuracy rewards used here. The length reward introduced in this work is also a known concept from existing RL literature.


W2) Lack of cost analysis: Although this work leverages inference-time scaling (extended reasoning and parallel sampling) to enhance performance, the accuracy/cost trade-offs introduced by scaling are not discussed. Moreover, the paper lacks cost comparison with existing methods, an important aspect for practical deployment in Text-to-SQL systems.


W3) BIRD's test set performance report: Reporting the performance on the private BIRD's test set is a conventional standard for fair comparison in the Text-to-SQL domain. Without the report on the BIRD test set or the confidence intervals of the reported results, it would be hard to assess the results against existing methods.

---

> ### Author Rebuttal · Authors · 2025-07-31
>
> Dear Reviewer,
>
> We sincerely thank you for your detailed review and constructive feedback. We will carefully address each of your questions and concerns with detailed clarifications and responses.
>
> **1. Distinction from Reasoning-SQL and Originality of SQL-R1 (In response to W1)**
>
> * **Our Response**: Thank you for your questions and suggestions for novelty。 **We need to unequivocally clarify that SQL-R1 and Reasoning-SQL are concurrent works**, and we publicly released our research in early April. It is crucial to acknowledge that Reasoning-SQL should not diminish our novelty or impose a requirement for us to compare ourselves with Reasoning-SQL.
>
> * **Additionally, our work is also significantly different from Reasoning-SQL**: first, our Reward function design is not consistent, and we achieve better execution accuracy without the complications of Schema filtering. Furthermore, we take our work seriously by training SQL-R1 entirely on synthetic data, which sets it apart from Reasoning-SQL that relies on the training set associated with BIRD and Spider. Our method demonstrates much stronger generalization capabilities. These fundamental differences undeniably confirm that SQL-R1 stands as an independent contribution, not a mere re-implementation.
>
> * In summary, SQL-R1 represents a novel contribution with its unique reward design and synthetic data training approach. While developed concurrently with Reasoning-SQL, our open-source implementation and distinct methodology demonstrate SQL-R1's independent value and advancement of NL2SQL capabilities.
>
>
>
> **2. Issue of the Experimental Results (In response to Q2 and W3)**
>
> * **Our Response**: Thank you for your question. For Q2, we expound relevant results in accordance with the article report of Reasoning-SQL.  We take this matter very seriously and feel it is crucial to address it transparently.  Therefore, we open source the code of SQL-R1, which is also the originality of our work at the experimental level.  In contrast, Reasoning-SQL does not open source the relevant code, which significantly complicates our ability to directly verify the experimental results of its report.  As a result, we adhere strictly to the original description and results provided in the article.
>
> * For W3, we are preparing to submit the results, but the relevant tests need to wait for the official conduct. According to the existing results, our confidence interval of the reported results is in [70,73]. Relevant results are added to the revised version when they are ready.
>
> **3. Latency of SQL-R1 with Self-Consistency (In response to W2)**
>
> * **Our Response**: This is an excellent suggestion for the cost. We evaluated the latency of SQL-R1 on the BIRD development set with the same inference setting mentioned in the Section 3.1:
>
> | Method | Candidate Selection | n | Latency (s) / Question | Total Tokens (K) / Question  | EX (%) |
> |--------|--------|---|------------------|--------| --------|
> | SQL-R1-7B | Greedy Search | 1 | 0.4 | 3.1 |  63.7 |
> | SQL-R1-7B | Self-Consistency | 8 | 1.1 | 23.1 |  66.6 |
> | CHESS | -| - | 251.3 | 320.8 |  61.5 |
>
> * Compared with the greedy search method (1 candidate and temperature=0), the self-consistency method with 8 candidates adds only an average of 0.7 seconds on each query while improving execution accuracy by 2.9%, which represents a favorable trade-off. Additionally, the self-consistency method employed in the proposed SQL-R1 is consistent with that of end-to-end NL2SQL models.
> * In comparison to CHESS (61.5%), which employs a multi-component workflow, the end-to-end SQL-R1 (66.6%) demonstrates not only superior execution accuracy but also significant advantages in latency and token usage efficiency. We will add these crucial comparative results to the revised manuscript.
>
> **4. Issue of Figure 9 (In response to Q1)**
>
> * **Our Response**: We appreciate your question. The results in Figure 9 are based on the 7B model tested without SFT. In the experiment section, we mentioned that SQL-R1-7B performs relatively well after SFT + RL, so its highest execution accuracy reaches 66.6. However, in the ablation experiment and the rest of the control, we used the model without SFT cold start for the convenience of comparison, so t. So, the highest accuracy of SQL-R1-7B here was 63.1, which was directly trained by RL. We will add these crucial comparative results to the revised manuscript.
>
>
> **5. Discussion of the reward function design (In response to Q3)**
>
> * **Our Response**: We appreciate your question. The weight of the reward function is designed by integrating its importance to NL2SQL and the perspective of preventing reward hacking. For the mentioned w/o $S_f$ ablation experiment, we modified the corresponding logic so that the model directly evaluated the executability, which easily led to execution errors when the model could not correctly extract SQL, thus obtaining invalid negative feedback and reducing the performance due to the impact of the sr function problem. We explain that this reward is sparse, and when facing more difficult samples in training, the model has less expectation to obtain this reward successfully. Hence, its influence on the model is relatively small. This is also an area for improvement, but it has undeniably contributed to the growth of the model's capabilities.
>
>
> **6. Discussion of length reward (In response to Q4)**
>
> * **Our Response**: We appreciate your question. The length reward is a smoothness reward designed to encourage the model to think deeper and try different ways of thinking. Figures 3 and 4 are good examples of qualitative comparison. The length reward helps the model navigate longer thought paths, resulting in a top-down thought process. Without the length reward, the model's ability to generate difficult samples will be weakened.
>
>
> We sincerely appreciate all the reviewers' constructive feedback and questions. We have provided detailed responses to address the concerns about SQL-R1's originality, experimental results, latency analysis, and reward function design. We have demonstrated that SQL-R1 is a concurrent and independent work with unique contributions, including novel reward design and a synthetic data training approach. The comprehensive experimental results and analyses validate the effectiveness and efficiency of our method. We are committed to maintaining transparency by open-sourcing our code and will continue improving our work based on the valuable suggestions received. These clarifications and additional analyses strengthen our paper's contribution to the NL2SQL research community.
>
>
>
>
> Sincerely,
>
> The Authors

---

> > ### Comment · Reviewer_Bp1S · 2025-08-06
> > **Rebuttal comments**
> >
> > I thank the authors for the effort they put into addressing my questions. I have read their responses to both my comments and those of the other reviewers.
> >
> > Below, I outline several unresolved concerns regarding the work:
> >
> > - I acknowledge that, according to NeurIPS policy, Reasoning-SQL and SQL-R1 would be considered concurrent works. However, as also noted by other reviewers, the algorithmic novelty of this submission remains limited. The authors identify two main contributions: (1) the use of execution accuracy as a task-specific reward, and (2) the incorporation of synthetic data during training. From an algorithmic standpoint, both techniques have already been explored in prior code generation literature. For example, regarding (1), the RLEF paper [1] demonstrated that using code execution as feedback can effectively guide reinforcement learning. Regarding (2), the synthetic data used in this work is sourced from [2] and is not a novel contribution of the current paper. Therefore, the proposed training methodology does not present a clear algorithmic distinction from standard fine-tuning practices.
> >
> > - In response to Q4: While Figures 3 and 4 provide qualitative examples comparing the model's outputs with and without RL training, they do not isolate or analyze the specific contribution of the length reward. Although I agree with the authors that longer chains of thought (CoT) may improve reasoning ability, the particular effect of the length reward remains unexplored and would benefit from ablation.
> >
> > - Regarding the cost analysis (W2): I appreciate the authors' clarification concerning token count and latency for SQL-R1. However, I would like to reiterate that comparing SQL-R1 to CHESS may not offer a meaningful benchmark. As noted by the authors, CHESS is a multi-component workflow based on older-generation models with significantly lower base performance than the model used in SQL-R1. Consequently, any performance gains may primarily result from the stronger base model rather than the proposed post-training approach. A more informative cost analysis could be achieved by adding comparisons in two dimensions: (1) base models (e.g., comparisons against state-of-the-art base models), and (2) Text-to-SQL systems built on the same base model (e.g., XiYan-SQL, CHASE-SQL).
> >
> > [1] *RLEF: Grounding Code LLMs in Execution Feedback with Reinforcement Learning*
> > [2] *Omnisql: Synthesizing High-Quality Text-to-SQL Data at Scale*

---

> > > ### Author Response · Authors · 2025-08-06
> > >
> > > In response to your inquiries, we would like to address each point as follows:
> > >
> > > **1. Further Discussion of Algorithm Novelty**
> > >
> > > We wish to clarify that the core innovation of our work does not lie solely in the isolated introduction of “execution accuracy as a reward” or “utilization of synthetic data.” Rather, it constitutes the **first proposal and validation of a holistic work specifically designed for complex NL2SQL tasks that integrates a multidimensional reward mechanism with a fully synthetic data based RL training paradigm**. The uniqueness and contribution of this work are reflected in the following two aspects:
> > >
> > > * **Differentiation from General Code Generation Literature**: The RLEF [1] you mentioned has achieved success in the realm of general code generation. However, the notion of “correctness” in the NL2SQL task is substantially more complex than the “executability” of general code. The execution feedback for general code is typically binary (success/failure), while our reward function is a **multidimensional and multilevel composite function specifically tailored for NL2SQL tasks**.It not only assesses the executability of the SQL but, more crucially, evaluates semantic accuracy by comparing the execution results with the ground truth. This is further enhanced by considering multiple dimensions, such as SQL query complexity and reasoning chain length.This meticulously designed reward structure aims to guide the model in performing deep reasoning within complex database relationships, thereby avoiding the generation of “shortcut” SQL queries that are syntaxically correct and executable but logically flawed.
> > >
> > > * **Unique Application Value of Synthetic Data**: We explicitly noted in our paper the derivation of synthetic data from OmniSQL [2], recognizing it as a solid foundation for our work.  Our contribution does not lie in data generation per se, but rather in the **first validation that, solely using a few synthetic data (5K), SQL-R1 can achieve competitive performance on unseen and complex real-world NL2SQL benchmarksunder zero-shot conditions**. This sharply contrasts with previous worksthat relied on costly, human-annotated in-domain training data from benchmarks such as BIRD and Spider. Our approach demonstrates not only superior generalization capabilities but also provides a **scalable and cost-effective new paradigm for solving NL2SQL tasks**, significantly reducing dependence on task-specific annotated data.
> > >
> > > Therefore, we believe that merely categorizing our training work as “standard fine-tuning practice” fails to capture its essence fully.
> > >
> > > **2. Contribution Analysis of Length Reward**
> > >
> > > * We conceptualize the Length Reward as both a **regularization term and an exploration incentive**, rather than a primary driver of performance. Its core function during training is to **encourage the model to generate longer and more in-depth chains of reasoning, thereby avoiding the pitfalls associated with "shortcut reasoning" that can lead to local optima**. That is supported by our qualitative analyses (Figures 3 and 4).
> > >
> > > * The role of the Length Reward is **auxiliary and synergistic**; it operates alongside primary reward objectives, such as execution accuracy, to guide the model in receiving positive feedback while exploring more complex reasoning pathways, particularly during the initial phases of training or when confronted with challenging problems. The absence of this reward mechanism would likely diminish the model's inclination to generate high-quality reasoning process, thereby constraining its capacity to tackle complex issues effectively.
> > >
> > > **3. Rebuttal Regarding Cost Analysis Benchmark Comparisons**
> > >
> > > * Our comparison with CHESS primarily aims to demonstrate the advantages of our **End-to-End approach** over the traditional **Multi-Component Workflow** in terms of **architectural paradigms**. We will clarify in the revised manuscript that the emphasis of this comparison.
> > >
> > > * To provide a more direct and equitable proof of performance gains, we will further demonstrate performance data comparisons under the same settings (Greedy Search) based on the aforementioned table. However, **it is essential to note that CHASE-SQL and Reasoning-SQL have not published their code, making it impossible for us to reproduce their reported results. Furthermore, CHASE-SQL relies on large-scale proprietary models such as Gemini 1.5 and Claude 3.5 Sonnet as base models, which renders a fair and reproducible comparison virtually impossible**. We firmly believe that reproducibility and openness are foundational pillars for advancing the field.
> > >
> > > | Method | Base Model | Latency (s) / Question | Total Tokens (K) / Question  | EX (%) |
> > > |--------|--------|---|------------------|--------|
> > > | Baseline | Qwen2.5-Coder-7B | 0.25 | 2.5 | 58.2 |
> > > | SQL-R1-7B | Qwen2.5-Coder-7B | **0.4** | **3.1** | **63.7** |
> > > | CHESS | - | 251.3 | 320.8 | 61.5 |
> > > | XiYanSQL-QwenCoder-7B-2504 | Qwen2.5-Coder-7B | 0.5 | 4.1 | 62.1 |

---

> ### Author Response · Authors · 2025-08-06
>
> Dear Reviewer Bp1S,
>
> We hope this message finds you well. As the deadline for discussions is fast approaching, we want to take a moment to express our sincere gratitude for your valuable review on our submission. Your insights are immensely appreciated, and we strive to ensure that our response has effectively addressed all your comments and major concerns.
>
> If you could confirm whether our rebuttal have met your expectations or if there are further aspects you would like us to clarify, it would be tremendously helpful.
>
> Thank you once again for your time and expertise. We look forward to hearing from you soon.
>
> Sincerely,
>
> The Authors

---

### Official Review · Reviewer_TvCP · 2025-06-29

**Clarity:** 3
**Significance:** 2
**Originality:** 2
**Rating:** 4
**Confidence:** 4

**Summary:**

This paper introduces SQL-R1, a novel NL2SQL model trained with reinforcement learning to improve reasoning and generalization in complex SQL generation tasks. By leveraging the GRPO algorithm and a structured reward function incorporating format, execution, result correctness, and reasoning trace quality, SQL-R1 produces both accurate and interpretable SQL outputs. The model is trained on synthetic data (SynSQL-2.5M) with a cold-start SFT phase, followed by RL fine-tuning. Experimental results on Spider and BIRD benchmarks show that SQL-R1 achieves competitive performance with closed-source models like GPT-4 while offering improved transparency and efficiency, especially on smaller base models.

**Questions:**

1.Spider dataset is an old dataset which could have contamination in pretrained models, why don't you use the Spider-2 dataset?
2. In the abstract, it's claimed that generalization is better with RL training, where can we tell there is generalization improvement?

**Ethical Concerns:**

["NO or VERY MINOR ethics concerns only"]

**Final Justification:**

The additional experimental results look good especially on more challenging spider 2 dataset. Error analysis by complexity is helpful to  justify the generalization capability.

**Limitations:**

Yes.

**Quality:**

3

**Strengths And Weaknesses:**

Strengths
- Clear Motivation for RL in NL2SQL: The paper is well-motivated by the limitations of SFT, especially in generalization and interpretability, - which are crucial in high-stakes domains like healthcare and finance.
- Custom Reward Design: The reward function, combining execution success, result correctness, formatting, and length penalties/rewards, is thoughtful and tailored to the NL2SQL task.
- Strong Benchmarking: SQL-R1 is evaluated across multiple model sizes and datasets (Spider, BIRD), showing competitive results against state-of-the-art systems, including GPT-4 and Gemini, especially considering it uses open-source LLMs.

Weaknesses
- Marginal Algorithmic Novelty: While the application of GRPO to NL2SQL is novel, the reinforcement learning framework itself is reused from prior work (e.g., DeepSeek-R1) with limited modification. The paper would benefit from a clearer discussion of what is novel versus reused.
- Limited Analysis of Errors, Generalization and Reward shaping: The paper does not sufficiently analyze where the model fails (e.g., complex joins, ambiguous queries), nor does it test generalization on other tasks. Ablations are done with reward shaping techniques, but  not enough analysis is given, i.e. how sensitive is the reward design. No error breakdown on failure cases, i.e. harder SQL generation improvement is bigger etc?

---

> ### Author Rebuttal · Authors · 2025-07-31
>
> Dear Reviewer,
>
> We appreciate your detailed review and constructive feedback. Your positive assessment and valuable suggestions will help enhance our paper's clarity.
>
> **1. Evaluation on Spider2.0 and generalisation evidence (In response to Question 1 and Question 2)**
>
> * **Our Response**: Thanks for the crucial questions. To address these concerns, we firstly evaluate SQL-R1 on **Spider 2.0-SQLite subset** . All models use the same self-consistency setup (n=8).
>
> | Model | Candidate Selection | Execution Accuracy (%) |
> |--------|--------|--------|
> | SQL-R1-7B | Self-Consistency  | **20.0** |
> | OmniSQL-7B [1] | Self-Consistency | 10.4 |
> | Qwen2.5-Coder-7B [1] | Self-Consistency | 2.2 |
> | Deepseek-V3 [1] | Self-Consistency | 15.6 |
> | GPT-4o [1] | Self-Consistency | 15.6 |
>
>
> * The findings presented above demonstrate that SQL-R1 surpasses existing methodologies in addressing Spider2.0 complex tasks related to NL2SQL.  This advantage can be ascribed to the model's enhanced inferential capabilities, which contribute to its overall efficacy in conducting sophisticated scenario analyses.  Regarding variations in dialects such as PostgreSQL and MySQL, additional time is required to perform thorough testing on Spider2.0.  We assure that supplementary results will be provided once they are finalized.
>
> * For generalisation of the SQL-R1, Table 4 shows execution accuracy on different NL2SQL tasks, including Spider-Realistic, Spider-DK and Spider-Syn.
>
> | NL2SQL Method | Base Model | Spider-DK | Spider-Syn | Spider-Realistic |
> |---------------|------------|-----------|------------|------------------|
> | SENSE | CodeLlama-7B | 77.9 | 72.6 | 82.7 |
> | ROUTE | Llama3-8B | 74.6 | 77.4 | 80.9 |
> | SQL-o1 | Llama3-8B | 78.7 | 72.6 | 82.7 |
> | OmniSQL | Qwen2.5-Coder-7B | 77.8 | 69.6 | 78.0 |
> | SQL-PaLM | PaLM-2 | 67.5 | 70.9 | 77.4 |
> | PURPLE | GPT-4 | 75.3 | 74.0 | 79.9 |
> | SQL-R1 (Ours) | Qwen2.5-Coder-3B | 70.5 | 66.4 | 71.5 |
> | SQL-R1 (Ours) | Qwen2.5-Coder-7B | 78.1 | 76.7 | 83.3 |
> | SQL-R1 (Ours) | Qwen2.5-Coder-14B | 79.3 | 78.5 | 86.2 |
>
> * The findings presented above demonstrate that SQL-R1 surpasses existing methodologies in addressing Spider2.0 complex tasks related to NL2SQL.  This advantage can be ascribed to the model's enhanced inferential capabilities, which contribute to its overall efficacy in conducting sophisticated scenario analyses. We will add these detailed explanations to the revised manuscript.
>
>
> **2. Algorithmic Novelty (In response to Weakness 1)**
>
> * **Our Response**: We appreciate your question. We now provide a more detailed explanation of the algorithmic novelty versus reuse. We acknowledge that our work draws inspiration from DeepSeek-R1, which is also mentioned in the Related Work section. However, we clarify that some task-specific redesigns presented here are novel and essential for advancing the framework.
>
>     1.  We have adapted the GRPO algorithm for the NL2SQL task, introducing a unique combination of reward functions specifically designed for this application, including execution and result rewards. This innovative approach enhances the model’s ability to generate accurate SQL queries from natural language input. We have customized and modified some task-specific principles (e.g, schema linking, database content retrieval) to better align with the specific requirements and challenges of the NL2SQL task.
>
>     2.  In this work, we utilized limited synthesized complex samples as training inputs for reinforcement learning (RL) to encourage the model to autonomously explore more robust reasoning pathways and inferential capabilities in NL2SQL translation.  This approach contrasts with the conventional methodologies that typically employ training sets derived from specific domain databases, such as the Spider and BIRD datasets.  Our findings indicate that synthesized data from diverse sources can significantly enhance the model's inferential strength and generalization ability, thereby contributing to the advancement of NL2SQL systems.
>
> * We hope that these explanations will help to address your concerns. And
>
> **3. Error Analysis by Complexity and Reward Sensitivity (In response to Weakness 2)**
>
> * **Our Response**: We sincerely appreciate your insightful question regarding the performance analysis across different complexity levels.
>
> * **Complexity Analysis**: We conducted a statistical analysis of the accuracy rates for various difficulty levels based on the BIRD development dataset, as presented in the table below:
>
> | Method | Simple | Moderate | Challenging | All |
> |---|---|---|---|---|
> | SQL-R1 (7B) | 72.1 | 60.8 | 51.0 | 66.6 |
> | SQL-R1 (14B) | 72.4 | 59.7 | 56.5 | 67.1 |
> | SuperSQL | 66.9 | 46.5 | 43.8 | 58.5 |
> | DAIL-SQL | 63.0 | 45.6 | 43.1 | 55.9 |
>
> - The results displayed show that SQL-R1 maintains a notable accuracy advantage over the current baseline across the three subsets: Simple, Moderate, and Challenging. This observation indicates that SQL-R1 exhibits a universally beneficial capability in reasoning and generation across different levels of SQL generation tasks.
>
> - Moreover, when we expanded SQL-R1 from a parameter size of 7B to 14B, the EX rates for the Simple and Moderate difficulty levels experienced a marginal change of only 0.3% and a decrease of 1.1%, respectively. However, it is essential to highlight the significant improvement in the Challenging difficulty category, where the accuracy surged from 51.0% to 56.5%, with an increase of 5.5 percentage points. This finding suggests that augmenting model capacity primarily alleviates the generalization bottleneck associated with high-difficulty queries.
>
> - This phenomenon corroborates previous research findings, which assert that larger-scale models are more adept at capturing long-range dependencies, particularly when queries involve multiple table joins, nested subqueries, or complex aggregations. The above insights underscore the critical role of model size of SQL-R1 in enhancing performance on intricate SQL generation tasks.
>
>
> * **Reward Sensitivity Analysis**: We conducted experiments on the BIRD development set by adjusting the weights of different reward components to evaluate their impact on model performance, using execution accuracy as the evaluation metric.
>
> | Reward Type | Set to 1.0 | Increase by 50% |
> |-------------|----------------------|------------------------|
> | Execution Reward | 60.0 | - |
> | Result Reward | 61.3 | - |
> | Length Reward | 59.7 | 60.6 |
>
> * The sensitivity analysis results reveal several key insights about the reward function design:
>
>     1.  Setting Execution and Result rewards to minimal weights leads to significant performance degradation, indicating potential reward hacking behavior where the model learns to exploit simpler patterns rather than developing robust reasoning capabilities.
>
>     2.  Increasing reward weights by large margins destabilizes training, particularly during early policy exploration phases.  The model may receive extreme rewards too frequently before establishing stable reasoning patterns.
>
>     3.  The lower sensitivity to Result reward adjustments suggests that while the model successfully explores executable SQL queries, semantic gaps between questions persist.  This makes it challenging for the model to achieve high Result rewards, even with increased weights consistently.
>
>     4.  A fixed Length reward proves suboptimal for encouraging deep reasoning.  The smooth sensitivity design we adopted maintains meaningful differentiation while avoiding training oscillations.  However, excessive Length rewards can also trigger reward hacking, where the model generates unnecessarily verbose solutions.
>
> * These findings validate our balanced reward function design choices that promote stable learning while preventing various forms of reward exploitation.  The results demonstrate the importance of careful reward calibration in reinforcement learning for complex NL2SQL tasks.
>
>
>
> We sincerely thank the reviewer for the constructive feedback and suggestions. We have provided detailed responses to address your concerns about algorithmic novelty, error analysis, and experimental results. We will incorporate all these clarifications and additional analyses in the revised manuscript.
>
> Sincerely,
>
> The Authors

---

### Official Review · Reviewer_FhNg · 2025-07-03

**Clarity:** 3
**Significance:** 2
**Originality:** 3
**Rating:** 5
**Confidence:** 4

**Summary:**

The paper proposes SQL-R1, an open-source NL2SQL system that fine-tunes a family of Qwen2.5-Coder models with GRPO. The main idea is to move beyond supervised fine-tuning by designing an RL reward that combines format, executability, result correctness and answer length. With a 7 B backbone, SQL-R1 reaches 88.7 % execution accuracy on Spider-Test and 66.6% on BIRD, outperforming other open LLMs and approaching closed models such as GPT-4o.

**Questions:**

How much slower is the self-consistency approach used to get the performance number? Is it the same approach across baselines?

**Ethical Concerns:**

["NO or VERY MINOR ethics concerns only"]

**Final Justification:**

Although this paper may not be very novel technically, I stick to my original recommendation of acceptance.

**Limitations:**

yes

**Quality:**

3

**Strengths And Weaknesses:**

### Strengths
1. This paper is among the first attempts to adopt GRPO for Text2SQL and has a clear layered reward definition that are well-motivated.
2. Large margins over prior open models on Spider and solid BIRD gains, achieved with modest RL data.
3. The results demonstrate that a 7B model trained with RL can match or beat much larger supervised models.
4. Good ablation study. Also the examples are clear in the paper, with interpretability in the reasoning process.

### Weaknesses
1. Experiments only cover SQLite; claim of cross-dialect robustness is untested — can authors provide PostgreSQL / MySQL results? Datasets like [Spider 2.0](https://spider2-sql.github.io/) might be a good add.
2. Eight-candidate self-consistency improves accuracy but may raise inference latency. Concrete numbers would help.
3. Spider and BIRD are human-curated, but training is wholly synthetic; real-world noise may hurt the performance. I'd like to see more discussion on robustness and transfer to very different domains, or multi-table queries.
4. The paper's novelty is not very significant, as it applies GRPO to a specific task.

---

> ### Author Rebuttal · Authors · 2025-07-31
>
> Dear Reviewer,
>
> We appreciate your detailed review and constructive feedback. Your positive assessment and valuable suggestions will help enhance our paper's clarity.
>
> **1. Latency of SQL-R1 with Self-Consistency (In response to Question and Weakness 2)**
>
> * **Our Response**: This is an excellent suggestion for the cost. We evaluated the latency of SQL-R1 on the BIRD development set, with the same inference setting and evaluation metric (Execution Accuracy, EX) mentioned in Section 3.1:
>
> | Method | Candidate Selection | n | Latency (s) / Question | Total Tokens (K) / Question  | Execution Accuracy (%) |
> |--------|--------|---|------------------|--------| --------|
> | SQL-R1-7B | Greedy Search | 1 | 0.4 | 3.1 |  63.7 |
> | SQL-R1-7B | Self-Consistency | 8 | 1.1 | 23.1 |  66.6 |
> | CHESS | -| - | 251.3 | 320.8 |  61.5 |
>
> * Compared with the greedy search method (1 candidate and temperature=0), the self-consistency method with 8 candidates adds only an average of 0.7 seconds on each query while improving execution accuracy by 2.9%, which represents a favorable trade-off. Additionally, the self-consistency method employed in the proposed SQL-R1 is consistent with that of end-to-end NL2SQL models.
> * In comparison to CHESS (61.5%), which employs a multi-component workflow, the end-to-end SQL-R1 (66.6%) demonstrates not only superior execution accuracy but also significant advantages in latency and token usage efficiency. We will add these crucial comparative results to the revised manuscript.
>
> **2. Robustness on Complex NL2SQL Tasks and Cross-dialect Generalization (In response to Weakness 1)**
> * **Our Response**: Thank you for suggesting the evaluation of SQL-R1 on the complex task related to NL2SQL. We now evaluated SQL-R1 on Spider2.0-SQLite subset, with the same inference setting and evaluation metric (Execution Accuracy, EX) mentioned in Section 3.1:
>
> | Model | Candidate Selection | Execution Accuracy (%) |
> |--------|--------|--------|
> | SQL-R1-7B | Self-Consistency  | **20.0** |
> | OmniSQL-7B [1] | Self-Consistency | 10.4 |
> | Qwen2.5-Coder-7B [1] | Self-Consistency | 2.2 |
> | Deepseek-V3 [1] | Self-Consistency | 15.6 |
> | GPT-4o [1] | Self-Consistency | 15.6 |
>
> * The results presented above demonstrate that SQL-R1 outperforms existing methodologies in handling complex NL2SQL tasks. This advantage can be attributed to the model's superior inference capabilities, which enhance its overall effectiveness in intricate scenario analyses.
> * As for different dialects like PostgreSQL and MySQL, we currently need more time to conduct comprehensive testing on Spider2.0. We assure that we will add more results in the revision version once they are ready.
>
> [1] Li, Haoyang, et al. "Omnisql: Synthesizing high-quality text-to-sql data at scale." arXiv preprint arXiv:2503.02240 (2025).
>
> **3. Robustness to Real-world Noise and Multi-table Generalization (In response to Weakness 3)**
>
> * **Our Response**: Thanks for your concern on the real-world noise. Table 4 presents evaluations conducted on the Spider-DK, Spider-Syn, and Spider-Realistic benchmarks, which include synthetic and real-world noise. We present a snapshot of the results in Table 4 below:
>
> | NL2SQL Method | Base Model | Spider-DK | Spider-Syn | Spider-Realistic |
> |---------------|------------|-----------|------------|------------------|
> | SENSE | CodeLlama-7B | 77.9 | 72.6 | 82.7 |
> | ROUTE | Llama3-8B | 74.6 | 77.4 | 80.9 |
> | SQL-o1 | Llama3-8B | 78.7 | 72.6 | 82.7 |
> | OmniSQL | Qwen2.5-Coder-7B | 77.8 | 69.6 | 78.0 |
> | SQL-PaLM | PaLM-2 | 67.5 | 70.9 | 77.4 |
> | PURPLE | GPT-4 | 75.3 | 74.0 | 79.9 |
> | SQL-R1 (Ours) | Qwen2.5-Coder-3B | 70.5 | 66.4 | 71.5 |
> | SQL-R1 (Ours) | Qwen2.5-Coder-7B | **78.1** | **76.7** | **83.3** |
> | SQL-R1 (Ours) | Qwen2.5-Coder-14B | **79.3** | **78.5** | **86.2** |
>
>
> * This results demonstrate that the methodology proposed in this paper consistently outperforms baseline methods while maintaining robust performance in challenging environments. Notably, SQL-R1 is trained entirely on synthetic data, yet the results in Table 1 and the aforementioned findings indicate its impressive resilience and effectiveness without the necessity for additional fine-tuning on other domain databases.
>
> * Additionally, we acknowledge that there remains room for improvement. For instance, exposing the model to more accurate filtered schemas could facilitate independent exploration and enhancement of its reasoning capabilities, particularly in complex table environments. Such approaches could lead to significant advancements in model performance when tackling diverse database contexts. Therefore, while our results are promising, ongoing research and experimentation in these areas could further elevate the robustness and applicability of our findings in real-world scenarios.
>
> **4. Novelty of SQL-R1 (In response to Weakness 4)**
>
> * **Our Response**: Thank you for your suggestion on the novelty of SQL-R1. We now provide a more detailed explanation of the novelty of SQL-R1.
>
>     * We have adapted the GRPO algorithm for the NL2SQL task, introducing a unique combination of reward functions specifically designed for this application, including execution and result rewards. This innovative approach enhances the model’s ability to generate accurate SQL queries from natural language input. We have customized and modified some task-specific principles (e.g, schema linking, database content retrieval) to better align with the specific requirements and challenges of the NL2SQL task.
>
>     * In this work, we utilized limited synthesized complex samples as training inputs for reinforcement learning (RL) to encourage the model to autonomously explore more robust reasoning pathways and inferential capabilities in NL2SQL translation.  This approach contrasts with the conventional methodologies that typically employ training sets derived from specific domain databases, such as the Spider and BIRD datasets.  Our findings indicate that synthesized data from diverse sources can significantly enhance the model's inferential strength and generalization ability, thereby contributing to the advancement of NL2SQL systems.
>
>
>
> We believe that these additions and discussions will greatly enhance the evaluation of the paper by providing essential comparative data and context regarding costs. We appreciate the reviewer’s constructive feedback, which aids us in improving the rigor of our study. We hope that the revised manuscript will meet the standards for acceptance.
>
> Sincerely,
>
> The Authors

---

> > ### Comment · Reviewer_TvCP · 2025-08-09
> >
> > The additional latency, robustness, and novelty analyses is very helpful. Thanks to the authors for such detailed responses. The self-consistency vs. greedy search latency numbers are convincing and show a favorable trade-off. The new Spider2.0-SQLite results are impressive and demonstrate SQL-R1’s ability to handle more complex NL2SQL tasks, though cross-dialect evaluation remains pending. Overall, the responses address my earlier concerns well, though I encourage including the cross-dialect results promised and ensuring that the latency/cost analysis is clearly integrated in the main text. I think my rating is reasonable. Thanks.

---

### Note · Authors · 2025-08-12

We sincerely thank all reviewers for their thorough evaluations and constructive feedback throughout the review process.

### Key Contributions

* SQL-R1 advances NL2SQL through **multidimensional reward mechanisms with synthetic data-based RL training**.  Our work introduces the first **composite reward function tailored for NL2SQL complexity**, evaluating SQL executability, semantic correctness, and reasoning depth.

* Our **synthetic data paradigm** demonstrates that using only 5K samples, SQL-R1 achieves competitive performance on unseen benchmarks, establishing a **scalable and cost-effective approach**.

### Experimental Validation

* **Performance Excellence**: SQL-R1-7B achieves 88.7% on Spider-Test and 66.6% on BIRD, matching larger proprietary models while maintaining superior efficiency (0.4s vs 251.3s latency).

* **Cross-Domain Generalization**: Testing on Spider-DK (78.1%), Spider-Syn (76.7%), Spider-Realistic (83.3%), and Spider2.0-SQLite (20.0% vs 15.6% for GPT-4o) demonstrates robust generalization.

**Complexity Benefits**: Analysis reveals SQL-R1's strength in challenging queries (5.5% improvement from 7B to 14B models), while maintaining consistent performance across difficulty levels.

### Addressing Concerns

* **Concurrent development with related work does not diminish independent innovation**.  Our unique combination of task-specific reward design, synthetic data utilization, and end-to-end architecture represents genuine methodological advancement.  The **multidimensional reward structure** addresses NL2SQL's semantic complexity challenges, ensuring logical correctness beyond simple executability.

* **Cost-effectiveness analysis** demonstrates practical advantages: SQL-R1's self-consistency adds only 0.7 seconds per query while improving accuracy by 2.9%, representing optimal trade-offs.

### Acknowledgments

* We appreciate all reviewers' constructive feedback.  Positive responses regarding experimental rigor and performance gains reinforce SQL-R1's contribution to open-source NL2SQL capabilities.  Critical questions helped us articulate our methodological innovations. Our open-source release ensures reproducibility and enables community advancement.  SQL-R1 effectively bridges the performance gap between open-source and proprietary models while maintaining superior efficiency and transparency.

---

### Decision · Program_Chairs · 2025-09-17

**Decision:**

Accept (poster)

**Comment:**

The goal of this paper is to fine-tune open LLMs to achieve sota performance on the task of translating NL to SQL queries and close the gap between the performance of open and closed models on this task. The authors fine-tune Qwen2.5-Coder to translate from NL to SQL using GRPO. They design an RL reward that combines format, executability, result correctness and answer length. Their model i.e. SQL-R1 reaches 88.7% execution accuracy on Spider-Test and 66.6% on BIRD, outperforming other open LLMs and approaching closed models such as GPT-4o. It also produces both accurate and interpretable SQL outputs. The model is trained on synthetic data (SynSQL-2.5M) with a cold-start SFT phase, followed by RL fine-tuning.

Strengths
- All reviewers are excited about the reward design, that takes many aspects of SQL queries into account, including the format. They find the reward clear and well motivated.
- All reviewers also appreciate the solid gains over baselines. Even their 7B model is competitive with larger closed models. They also appreciate the ablations in the paper. Overall, the paper is clear and strongly motivated, with a strong experimental section. The paper also studies the necessity of supervised fine-tuning as a prerequisite (cold start problem) and the requirement of high-quality training data, all of which are very interesting for this particular task.

Weaknesses
- The authors have addressed several of the weaknesses in their rebuttal:
    - Lack of results on spider 2.0, which the authors have provided.
    - Latency numbers for inference efficiency particularly in the self-consistency regime - the authors have resolved this.
    - Missing error analysis - the authors have provided this
    - Missing baselines of sql-o1 and alpha-sql - resolved!

- While 3/4 reviewers would like to accept the paper, what remains is two main weaknesses pointed out by one reviewer.

1. The lack of novelty i.e. the paper applies a known method (GRPO) to a known task. There are also existing papers that use execution feedback as a reward, so that part is not novel either. There is a lot of back and forth on this topic between the authors and the reviewers. The authors point out that while each individual aspect is not novel, the synergistic combination of multiple techniques is what makes the model work - and is non-trivial to achieve. They also look at semantic accuracy, and oppose the idea that this work is merely "fine-tuning".
2. Lack of results on BIRD's held out test set - which is missing in the paper, and the authors are waiting for the results from this submission.

Overall, with three accepts, this paper represents a holistic post-training method for the task of NL to SQL. This paper can be a solid reference manual as a strong recipe on how to post train a SQL generation model, serving as a strong baseline for any future NL -> SQL methods. I therefore recommend accepting this paper.